# Mechanical feedback and robustness of apical constrictions in *Drosophila* embryo ventral furrow formation

**Michael C. Holcomb**[1¤a], **Guo-Jie Jason Gao**[2], **Mahsa Servati**[1¤b], **Dylan Schneider**[3], **Presley K. McNeely**[1], **Jeffrey H. Thomas**[4]*, **Jerzy Blawzdziewicz**[1,3]*

**1** Department of Physics and Astronomy, Texas Tech University, Lubbock, Texas, United States of America,
**2** Department of Mathematical and Systems Engineering, Shizuoka University, Hamamatsu, Japan,
**3** Department of Mechanical Engineering, Texas Tech University, Lubbock, Texas, United States of America,
**4** Department of Cell Biology and Biochemistry, Texas Tech University Health Sciences Center, Lubbock, Texas, United States of America

¤a Current address: Department of Physics and Geosciences, Angelo State University, Texas, United States of America
¤b Current address: School of Health Sciences, Purdue University, West Lafayette, Indiana, United States of America
* jeffrey.thomas@ttuhsc.edu (JHT); jerzy.blawzdziewicz@ttu.edu (JB)

**Data Availability Statement:** All relevant experimental data are available from Dryad at https://doi.org/10.5061/dryad.m7q37nv. The numerical code is available from GitHub at

## Abstract

Formation of the ventral furrow in the *Drosophila* embryo relies on the apical constriction of cells in the ventral region to produce bending forces that drive tissue invagination. In our recent paper we observed that apical constrictions during the initial phase of ventral furrow formation produce elongated patterns of cellular constriction chains prior to invagination and argued that these are indicative of tensile stress feedback. Here, we quantitatively analyze the constriction patterns preceding ventral furrow formation and find that they are consistent with the predictions of our active-granular-fluid model of a monolayer of mechanically coupled stress-sensitive constricting particles. Our model shows that tensile feedback causes constriction chains to develop along underlying precursor tensile stress chains that gradually strengthen with subsequent cellular constrictions. As seen in both our model and available optogenetic experiments, this mechanism allows constriction chains to penetrate or circumvent zones of reduced cell contractility, thus increasing the robustness of ventral furrow formation to spatial variation of cell contractility by rescuing cellular constrictions in the disrupted regions.

## Author summary

Invagination of epithelial tissues is a common means by which living organisms generate their form and structure. Using the ventral furrow in the *Drosophila* embryo, our study addresses the fundamental question of how individual cells coordinate their activities to produce well-organized and robust invagination. Ventral furrow formation, the best studied example of epithelial invagination during early embryonic development, is initiated by constrictions of the apical (outer) surfaces of some of the cells in the epithelial cell layer to

https://github.com/Guo-Jie-Jason-Gao/Drosophila-Active-Granular-Fluid-Model.

**Funding:** JB was partially supported by the National Science Foundation (NSF Grant CBET 1603627). JHT received support from the South Plains Foundation and from Texas Tech University Health Sciences Center Department of Cell Biology and Biochemistry (USA). GJG acknowledges the computational facility made available by the startup funding of Shizuoka University (Japan). MCH was partially supported by the Texas Tech University (USA) Doctoral Dissertation Completion Fellowship and startup funding from Angelo State University (USA). The funders had no role in study design, data collection and analysis, decision to publish, or preparation of the manuscript.

**Competing interests:** The authors have declared that no competing interests exist.

produce bending forces that cause tissue buckling. We find that these apical constrictions follow a pattern based on underlying stress chains that run along the axis of invagination. Our analysis of the experimentally observed and numerically simulated constriction patterns shows that tensile stress prompts cells to constrict, and that constrictions strengthen the stress chains. This two-way interaction allows cells to coordinate their constrictions via mechanical feedback. Reduction of regional cellular contractility by both experimental and computational perturbations shows that stress chains penetrate the affected tissue, giving rise to cell constrictions. Thus, stress feedback increases the robustness of tissue invagination. Our study elucidates how cell communication via mechanical forces helps to achieve robust structure formation.

## Introduction

Previous research efforts to understand morphogenesis have primarily focused on the identification of genetic information and biochemical signals involved in the formation of embryonic architecture. In recent years, compelling evidence that cell communication via mechanical forces is crucial in orchestrating morphogenetic processes has emerged [1–10]. In *Drosophila* gastrulation, mechanical signaling has been shown to be a triggering mechanism for morphogenetic events [11–15], and mechanical feedback to be a factor in the coordination of cellular activities in the mesoderm primordium [5]. Mechanical feedback is also involved in remodeling subcellular components such as adherens junctions [15] and the supracellular actomyosin meshwork [16]. The earliest morphogenetic movement in *Drosophila* gastrulation is ventral furrow formation (VFF). During VFF the cells of the ventral mesoderm primordium, a region approximately 12 cells wide and 80 cells long, in *Drosophila* embryos are capable of mechanical activity due to expression of the regulatory genes *twist* and *snail* [13, 17–21], and are internalized through invagination of this epithelial tissue [17, 22].

VFF occurs in distinct steps: (1) apical flattening; (2) the early slow phase during which a growing number of mesoderm primordium cells undergo apical constrictions; (3) the fast transition phase, where under the influence of the Fog signal the remaining cells undergo apical constrictions at the onset of buckling of the cell layer; and (4) the invagination phase during which the ventral furrow forms. The shift from the slow phase of VFF to the fast phase and the subsequent invagination occurs when approximately 40% of cells in the mesoderm primordium have constricted [17, 22]. Early time-lapse studies show that the cells in the ventral mesoderm maintain their positions during the apical constriction phase [22, 23].

Apical constrictions in the slow phase, originally thought to be random, later were shown to be correlated with the apical constrictions of neighboring cells [22, 24]. In embryos, we noticed that correlation of apical constrictions extended beyond that of merely nearest neighboring cells to form well-defined correlated structures which we termed cellular constriction chains (CCCs). Formation of constriction chains implies the existence of strong spatial and directional correlations between the constriction events [5]. We suggested that CCCs originate from communication between cells via tensile stress that they exert on the surrounding tissue. Tension is likely carried through the supracellular actomyosin cables that function in mechanosensation [16] and are generally oriented along the anteroposterior axis.

According to our theoretical analysis, tensile stress builds up near chain ends and between linearly aligned chain fragments. Therefore, positive tensile-stress feedback results in the formation and growth of constriction chains. In our theoretical active granular fluid (AGF)

model, cells (more specifically, their mechanically active apical ends) are treated as mechanically sensitive interacting particles undergoing a stochastic constriction process.

Here, we test our model through detailed imaging and analyses of the slow phases of apical constriction during VFF, and observe the progressive development of CCCs *in vivo*. Using the AGF model combined with a novel stress-based Voronoi construction, we predict a number of qualitative and quantitative characteristics of the stress-guided apical constriction process, and compare them with the corresponding features of constriction patterns in live embryos. We find that experimental data agree well with our predictions for a system with tensile-stress feedback, and that these data are inconsistent with results for purely random uncorrelated constrictions or for only neighbor-correlated apical constrictions. The provided evidence thus indicates that tensile-stress feedback is an important controlling factor coordinating apical constrictions. Our AGF simulations demonstrate that coordination of apical constrictions by tensile stress allows CCCs to penetrate or bypass regions of reduced constriction probability, thus rescuing the constriction process. The rescue mechanism relies on a buildup of a strong precursor stress in the disrupted zones. An analysis of constriction patterns in live embryos in which contractile force transmission has been reduced locally using optogenetic techniques [25] is consistent with our theoretical findings and consistent with the recent findings of others [16, 26]. We thus conclude that mechanical feedback and the associated formation of cellular constriction chains reduce nonuniformities of the constriction pattern and aid robustness of the VFF in the presence of environmental or genetic fluctuations.

## Results

### Apical constriction-chain patterns develop during ventral-furrow formation

We imaged the ventral sides of live Spider-GFP *Drosophila* embryos to visualize apical cell membranes [27]. The ventral furrow forming region was imaged in embryos during early gastrulation before invagination of the ventral furrow (Fig 1). Imaging began before the commencement of apical constrictions and continued through the initial stage of VFF invagination (Fig 2). The initial timepoint was defined by the first persistent apical constriction.

We processed the images and identified the areas and major and minor axes of the ventral cells. Apical constrictions were defined by the length of the minor axis of each cell apex. These constrictions are typically anisotropic, resulting in the minor axis undergoing the most pronounced changes. This behavior is consistent with the dorsal-ventral (ventral-lateral) bias in apical constriction previously observed [28, 29]. Specifically, the cells are marked as constricted if the minor-axis reduction factor

$$r = \lambda/\lambda_{\mathrm{ref}} \tag{1}$$

falls below a specified threshold value $r_{\mathrm{c}}$

$$r \leq r_{\mathrm{c}}. \tag{2}$$

Here $\lambda$ is the minor axis length of a given cell apex, and $\lambda_{\mathrm{ref}}$ is the reference value established for each individual cell by averaging over twenty sequential frames before constrictions begin.

We note that constrictions of individual cells occur in a pulsatile, non-monotonic process, in which some pulses are unratcheted (reversible) and some are ratcheted (irreversible) [21, 24]. Therefore, there is no uniquely defined threshold value $r_{\mathrm{c}}$ to determine whether a cell is constricted. To reveal spatial and temporal correlations between constrictions of different strengths, developing constriction patterns were identified using several threshold values in

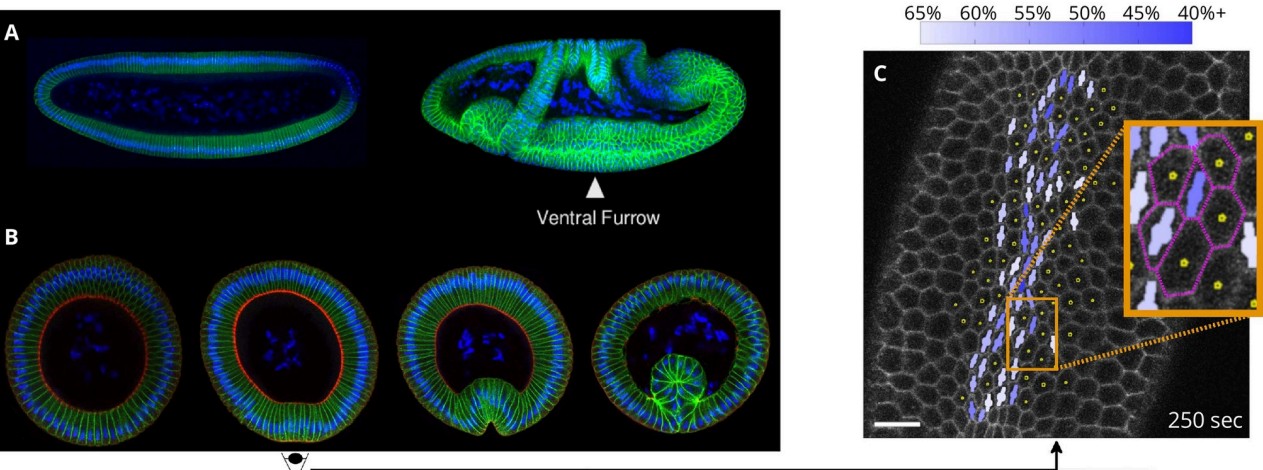

**Fig 1. Ventral furrow formation in the *Drosophila* embryo.** (**A**) The embryo before and after gastrulation; position of the ventral furrow is indicated. (**B**) Cross-sectional images of fixed embryos, showing the progression of VFF. The second image shows the stage imaged and analyzed. (**C**) Processed confocal microscopy image of a live Spider-GFP *Drosophila* embryo near the end of the initial slow apical constriction phase of VFF, showing apical cell constriction characteristics. Constricted cells are indicated by a bar with a circle (bar–circle), where the bar represents the direction and length of the major cell axis, and the size of the circle indicates the relative cell area. Color saturation specifies the degree of constriction, measured by the reduction of the cells minor axis length relative to the reference value (as defined in Eq 1 and indicated by the color bar). Tracked cells are indicated by dots. Scale bar: 10 μm. Embryos (A,B) were stained with Hoechst (blue) to visualize nuclei, antibodies to Neurotactin (green) to visualize cell membranes, and antibodies to Zipper (myosin heavy chain, red).

the range from $r_c = 0.6$ to $0.9$. Since strong apical constrictions are typically ratcheted, a strong cutoff (a small value of $r_c$) corresponds primarily to ratcheted constrictions. The constricted cells are indicated using a bar–circle symbol representing the major axis of the marked cell and the cell area (Fig 1C).

To visualize strong constrictions, cells are marked as constricted when the minor axis falls below 65% of the reference value ($r_c = 0.65$). At this cutoff, spatial correlations between constrictions are pronounced (Fig 2). The slow phase of VFF begins with constrictions of individual cells that gradually form a sparse pattern of singlets and doublets concentrated around the ventral-most region of the embryo (the sideline of the long-axis). After this initial stage, CCCs (connected chain-like arrangements of constricting cells) emerge and rapidly grow in length. Expanding constriction chains branch and merge, promoting interconnection of the CCC network and eventually leading to percolation of cellular-constriction chains across the active region.

Different threshold values of $r_c$ provide additional insights from an analysis of constriction patterns. Comparison of CCCs from a single embryo using different cutoffs shows that constriction chains visualized using a stronger cutoff are similar to those obtained with a weaker cutoff at an earlier time. Moreover, using a weaker cutoff shows more fragmented chains (Fig 3).

The above features of the constriction patterns imply that spatial correlations between apical constrictions emerge early during the apical constriction process and build up as constrictions progress. The enhancement of the constriction pattern over time occurs in a fluctuating manner, because of the existence of unratcheted constrictions. These patterns indicate that the collective apical constrictions involve significant spatial and directional correlations. We postulate that these chain patterns arise from mechanical feedback coordination of the constriction process.

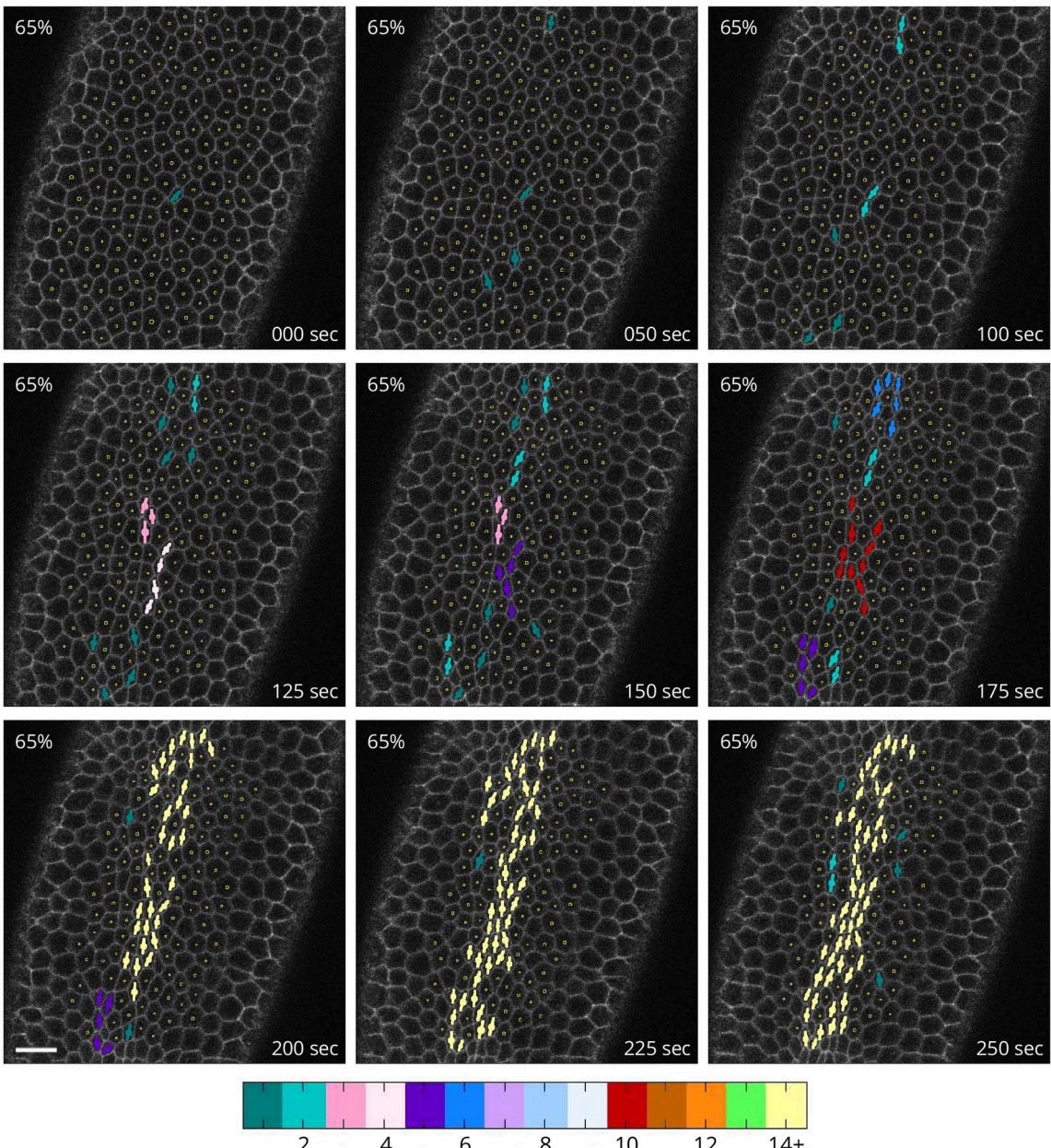

**Fig 2. Development of constriction chains during the slow apical constriction phase of VFF.** Time lapse images of the underside of a Spider-GFP embryo showing constricted apices defined by the minor-axis reduction threshold $r_c$ = 0.65. Symbols per Fig 1. Color bar indicates the number of cells in the constricted-cell cluster interconnected via shared neighbors. Constricted cells initially form a sparse pattern, extending in the anteroposterior direction. Subsequently, cells rapidly interconnect into elongated chains of constricted cells, forming a constricted-cell network that percolates across the mesoderm primordium along the anteroposterior axis. Many of the marked cells remain constricted across multiple frames, indicating that they are undergoing ratcheted constrictions [24]. Unratcheted constrictions can also be seen. Time point zero was determined by the first observed apical constriction. Scale bar: 10 μm.

## The epithelial blastoderm can be represented as a system of interacting active particles

To elucidate the origin of CCCs and investigate their effect on the robustness of VFF, we use the original AGF model with mechanical feedback [5] further developed here. Unlike more

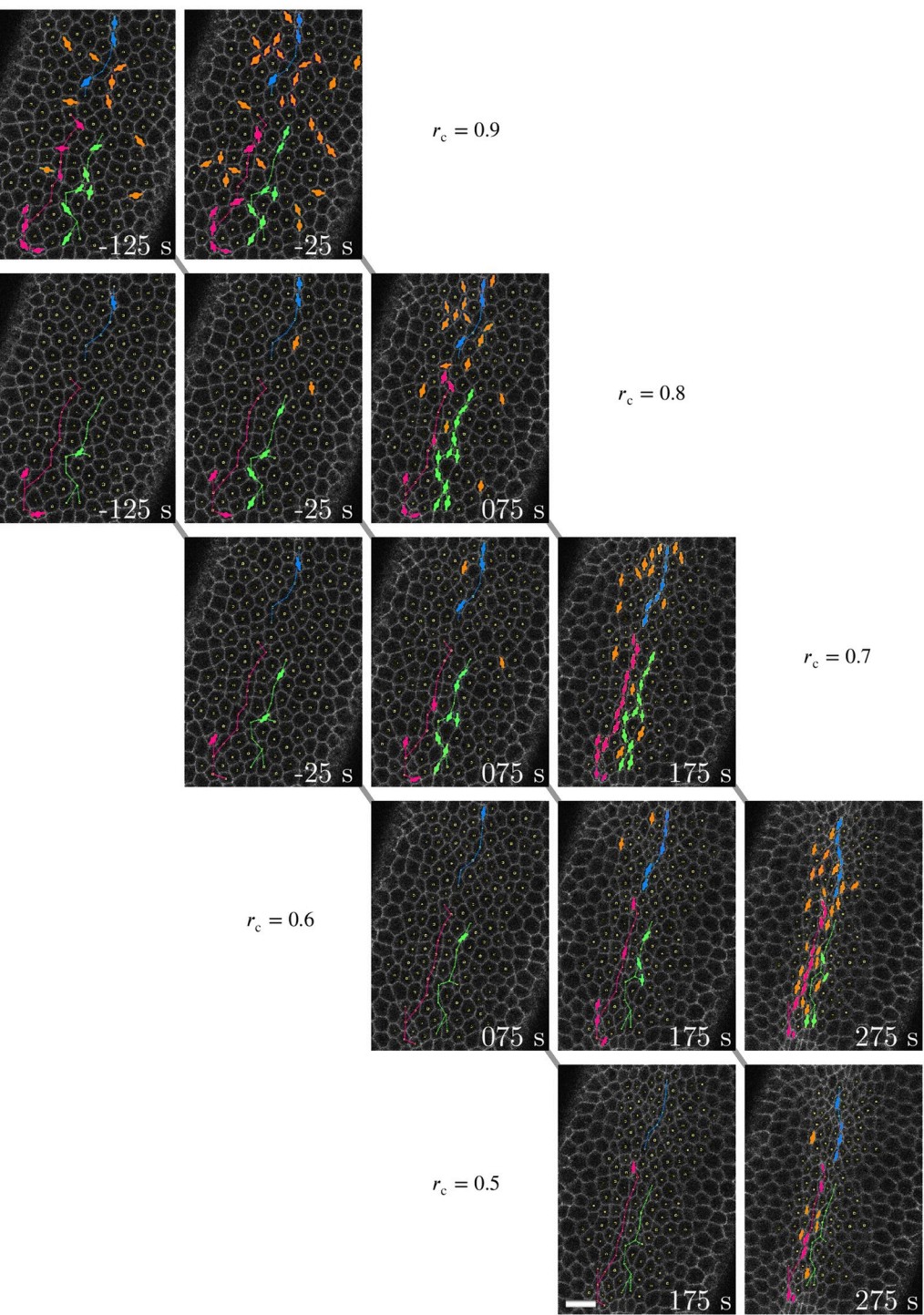

**Fig 3. Comparison of cellular-constriction chain progression with respect to time and degree of constriction.**
Processed images showing the apical constriction phase of VFF in a single Spider-GFP *Drosophila* embryo using different minor-axis reduction thresholds $r_c$ (as labeled). Symbols per Fig 2. Time-lapse sequences in different rows indicate that CCC development is similar between a stronger $r_c$ and a weaker $r_c$ at an earlier time. Notable chains that persist between different $r_c$ strengths are demarcated in either pink, green, or blue and indicated by thin lines. Scale bar: 10 μm.

common vertex models [30, 31], the AGF approach does not deal with individual cell membranes. Instead, entire apical cell ends are represented as mechanically coupled, stress-responsive active particles that are capable of random constrictions. This coarse-grained AGF technique can readily describe populations of cell membranes, adherens junctions, the cortical apical actomyosin meshwork, actomyosin rings at the cell junction, and supracellular actomyosin filaments [16, 21, 26]. While a vertex modeling approach could be applied, a particle-based model that treats cells as undivided entities described by their effective properties is more appropriate here because of its simplicity. Our model has only three dimensionless parameters ($p$, $f_c$, and $\beta$ described below) that control the constriction process, allowing straightforward interpretation of the results. (For a recent review of the applications of particle-based models to study tissue mechanics see Ref. [32]).

The AGF system consists of a band of active cells (yellow) surrounded by inactive cells (gray) (Fig 4). The active cells occupy a stripe approximately 12 cells wide and 80 cells long, corresponding to the size of the mesoderm primordium in the *Drosophila* embryo [22]. The cells are modeled as finite-size particles interacting via isotropic potential forces. Each cell is characterized by the effective diameter $d_i$, corresponding to the range of the repulsive part of the interaction potential. In Fig 4B, 4C, 4E and 4F this effective size is represented by a circle of the diameter $d_i$.

The active cells undergo a stochastic constriction process in which a particle $i$ can instantaneously constrict by reducing its effective diameter $d_i$ by a specified constriction factor $f_c$,

$$d_i \rightarrow f_c d_i. \tag{3}$$

Since we focus on ratcheted constrictions [24], we assume that already constricted cells (brown) do not unconstrict or undergo any other size changes.

The inactive particles occupy the region outside the mesoderm primordium. These do not have constriction activity, but they contribute to the mechanical environment for the constriction process. An outer portion of the inactive region is modeled as an effective elastic medium (Fig 4D).

The cells interact via elastic and adhesive forces approximated using a pairwise spring potential,

$$V_{ij} = \begin{cases} \dfrac{\epsilon}{2}(1 - r_{ij}/d_{ij})^2 & i,j \text{ connected neighbors} \\[2ex] \dfrac{\epsilon}{2}(1 - r_{ij}/d_{ij})^2 \Theta(1 - r_{ij}/d_{ij}) & i,j \text{ unconnected cells.} \end{cases} \tag{4}$$

Here $\epsilon$ is the characteristic energy scale, $r_{ij}$ is the separation between cells $i$ and $j$, and $d_{ij} = \frac{1}{2}(d_i + d_j)$ is their average diameter. The interaction potential has an attractive part ($r_{ij} > d_{ij}$) and a repulsive part ($r_{ij} < d_{ij}$). The Heaviside step function $\Theta(r)$ selects the repulsive part for particles that are not connected neighbors and therefore do not exhibit cell adhesion. During the constriction process unconnected cells typically are outside the repulsion range, but these interactions play a role during the preparation of the initial state (see Methods section).

The repulsive part of the potential Eq 4 mimics elastic interactions of deformed cells that are pressed together. The attractive part represents a combination of elastic and adhesive forces acting between cells that are connected by spot adherens junctions, when these cells are pulled apart. The intercellular interaction forces

$$f_{ij} = -\mathrm{d}V_{ij}/\mathrm{d}r_{ij}, \tag{5}$$

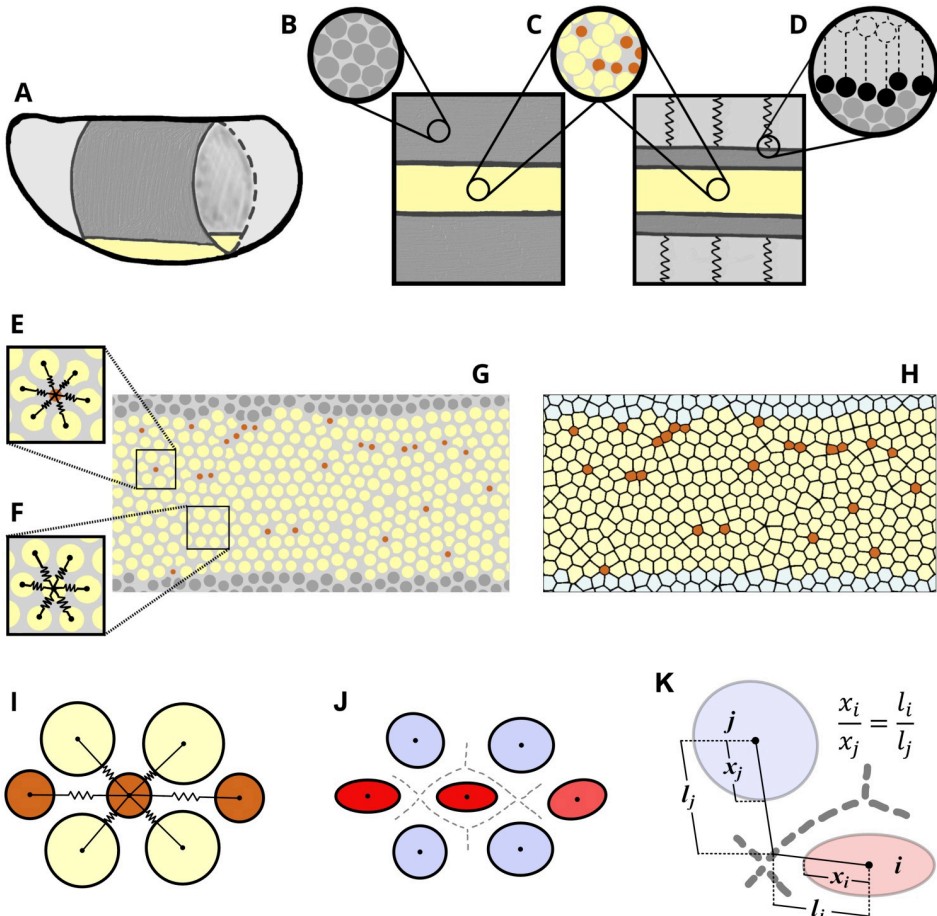

**Fig 4. AGF model schematic.** (**A**) The model represents the apical surface of the *Drosophila* embryo. The region of active cells that undergo apical constrictions during the initial phase of VFF in the mesoderm primordium is shown in yellow. Inactive cells in the lateral and dorsal regions (gray) do not experience constrictions. (**B,C**) Insets show how cells in both inactive (B) and active (C) regions are modeled as force centers represented by disks. (**D**) Inset shows how we implicitly model a portion of the inactive region as an effective elastic medium, in the interest of computational efficiency. (**E,F**) Cells interact with their adjacent neighbors through isotropic pairwise elastic potentials. Cells maintain the same neighbors even after constriction (constricted cells are marked in brown). (**G,H**) Particle configuration can either be represented by circles denoting force centers (G) or by Voronoi cells (H) generated using an anisotropic distance function. (**I,J**) The isotropic pairwise elastic potentials (I) are used to calculate the normalized virial stress tensor (Eq 10). A combination of an isotropic tensor (undeformed cell) and the virial stress tensor is used to define ellipses (J) representing a cell's degree of deformation (Eq 9). The higher the eccentricity, the more anisotropic the deformation. The fill color shows tensile (red) and compressive (blue) major stresses. (**K**) Our augmented Voronoi tessellation establishes cell boundaries such that the ratio of distances from adjacent cell centers matches the ratio of center to edge distances of their representative ellipses.

not only give the epithelial layer its mechanical integrity, but also provide feedback that coordinates apical constrictions.

The epithelial structure of the tissue is maintained throughout the VFF process. We mimic this by establishing a neighbor list of adjacent, connected particles from the initial configuration. This neighbor list is maintained throughout the simulation.

To describe a stress-correlated stochastic constriction process, we perform a sequence of simulation steps in which each active cell $i$ can constrict with the stress-dependent probability $P_i(s_i)$, where $s_i$ is the feedback parameter associated with tensile forces acting on the cell. The

constrictions are followed by mechanical relaxation of the medium to ensure that the process is quasistatic and therefore governed by the equilibrium stress distribution.

In our numerical model, the feedback parameter

$$s_i = (\sigma_i / \sigma_{\text{ref}})^p \Theta(\sigma_i) \tag{6}$$

is given in terms of the triggering stress exerted on cell *i* by its surrounding cells,

$$\sigma_i = -\epsilon^{-1} \sum_{j \neq i} d_{ij} f_{ij}. \tag{7}$$

The Heaviside step function $\Theta$ in the feedback parameter defined by Eq 6 selects the tensile-stress domain $\sigma_i > 0$, and *p* is the stress-sensitivity profile parameter. As in our previous study [5] we use $p = 3$, which corresponds to enhanced sensitivity to large stresses. The stress $\sigma_i$ in Eq 6 is normalized by the average tensile stress $\sigma_{\text{ref}}$ experienced by a single cell constricted in the initial configuration, obtained from full simulations of the entire system of 6,400 cells without the implicit mesoderm representation.

The overall tension is oriented along the anteroposterior axis during VFF [33]. Supracellular actomyosin cables are likewise connected by adherens junctions oriented along the anteroposterior axis and function in mechanosensation [16, 26, 34]. In our model the tension is also oriented along the anteroposterior axis, because of the imposed boundary conditions and asymmetry associated with the presence of the inactive region.

The constriction probabilities $P_i(s_i)$ are calculated according to the relation

$$P_i(s_i) = \frac{\alpha_i(1 + \beta s_i)}{N_a(1 + \beta)}, \tag{8}$$

where $N_a$ is the current number of unconstricted active cells. The normalization by $N_a$ ensures that $P_i \ll 1$ for the range of stresses that occur in our system and that approximately the same small number of cells constrict in each simulation step (typically one or two particles). Our model thus can be considered as a discretization of a continuous stochastic process in which constrictions occur instantaneously with a prescribed (stress-dependent) probability per unit time.

The coupling constant $\beta \geq 0$ determines the responsiveness of cellular constrictions to the stress feedback parameter $s_i$, with $\beta = 0$ corresponding to the stress insensitive (random) case, and $\beta \to \infty$ describing a system where tensile stress is required for constrictions. Throughout this paper we use $\beta = 250$ in our simulations of tensile-stress sensitive systems; this value of $\beta$ corresponds to a relatively strong stress coupling and provides results consistent with experimental data. The parameter $\alpha_i > 0$ determines the overall magnitude of the constriction probability of a given cell. In our simulations, we generally consider $\alpha_i = 1$ (the same value for all cells); however, in our robustness tests $\alpha_i$ is different for cells in different regions of mesoderm primordium. Fitting of our simulation data to experimental results was accomplished using a single fitting parameter, the coupling strength parameter $\beta$ as discussed in the Methods section.

## An augmented form of Voronoi tessellation can be used to construct a confluent cell layer

The circles in the schematic depicted in Fig 4 indicate the positions $\mathbf{r}_i$ and repulsive-potential ranges $d_i$ of the force centers defined by the potentials in Eq 4. However, the circles do not represent the actual cell shapes because cells are deformed through their interaction with other cells (as members in a confluent epithelial layer *in vivo*).

To evaluate cell shapes in a confluent cell layer described by the force center model, we introduce an augmented Voronoi construction that uses the force-center positions $\mathbf{r}_i$ and the interaction-force vectors $\mathbf{f}_{ij}$ to create a tessellation that is consistent with the system mechanics (Fig 4H). In previous studies [35–39], Voronoi tessellation was applied to infer approximate cell shapes from the positions of cell centers alone, without using any cell-shape related parameters. While this standard tessellation technique was shown to yield quite accurate representations of actual cell-shape distributions in some epithelial tissues [40], we find that it is inadequate for description of the apical constriction process. In particular, the standard technique significantly underestimates size differences between constricted and unconstricted cells and fails to predict cell elongation in the direction of the constriction chains, an important feature observed *in vivo* (Figs 1C, 2 and 3).

To overcome these difficulties, our augmented Voronoi tessellation is defined in terms of the cellular shape tensor (Fig 4I–4K)

$$\mathbf{D}_i = d_i(\mathbf{I} + s_0^{-1}\mathbf{S}_i), \tag{9}$$

determined from the normalized virial stress [41]

$$\mathbf{S}_i = -\frac{1}{2\epsilon}\sum_{j \neq i}\mathbf{r}_{ij}\mathbf{f}_{ij}, \tag{10}$$

where $\mathbf{r}_{ij} = \mathbf{r}_i - \mathbf{r}_j$ is the relative position of the force centers $\mathbf{r}_i$ and $\mathbf{r}_j$, and $\mathbf{f}_{ij} = f_{ij}\hat{\mathbf{r}}_{ij}$ is the force vector in the radial direction $\hat{\mathbf{r}}_{ij} = \mathbf{r}_{ij}/r_{ij}$. The scale factor $s_0$ is associated with a typical number of springs that contribute to cellular deformation in a given direction. We use $s_0 \approx 2$ in our simulations.

The shape tensor $\mathbf{D}_i$, Eq 9, defines anisotropic distance function associated with each cell. Specifically, the weighted distance $\bar{\rho}_i$ between the trial point $\boldsymbol{\rho}$ and particle representing cell $i$ is given by the formula

$$\bar{\rho}_i = \frac{\rho_i}{\hat{\boldsymbol{\rho}}_i \cdot \mathbf{D}_i \cdot \hat{\boldsymbol{\rho}}_i}, \tag{11}$$

where $\boldsymbol{\rho}_i = \boldsymbol{\rho} - \mathbf{r}_i$ is the relative particle–point position, $\rho_i = |\boldsymbol{\rho}_i|$ is the unweighted particle–point distance, and $\hat{\boldsymbol{\rho}}_i$ is the unit vector in the particle–point direction. To generate the augmented Voronoi tessellation based on the measure defined by Eq 11, each trial point $\boldsymbol{\rho}$ is assigned to a particle $i$ for which the weighted distance $\bar{\rho}_i$ has the smallest value. The Voronoi tessellation of the force-center system shown in Fig 4G is depicted in Fig 4H. We find that the augmented Voronoi algorithm renders a realistic representation of a confluent cellular medium with a significant degree of polydispersity (i.e. variety of cell sizes) and local anisotropy associated with the presence of CCCs.

## The active granular fluid model predicts that tensile feedback results in formation of cellular constriction chains

The AGF model predicts the essential role of tensile feedback in the formation of CCCs. Simulations of a system with a relatively strong feedback ($\beta = 250$) show chain-like formations of constrictions (Fig 5A), similar to CCCs observed *in vivo* (Fig 2). At the point where approximately 40% of cells have constricted, when the embryo would be completing the initial slower phase of VFF *in vivo* [22], the chains organize into a percolating network, formed through the interconnection of shorter chains that emerged during earlier stages of the process. Contrariwise, a combination of small clumps and short chains is predicted for stress-insensitive apical

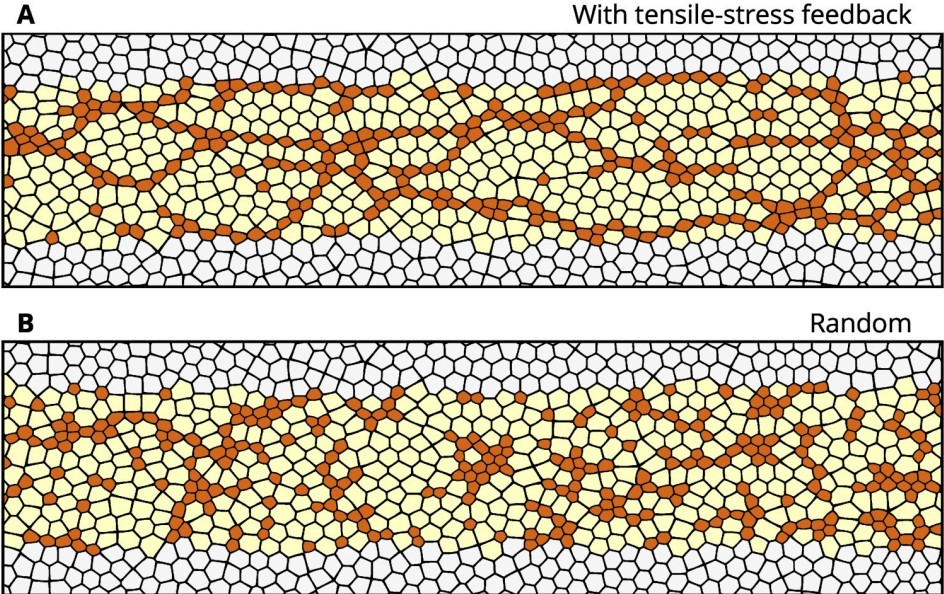

**Fig 5. Model showing the effect of tensile-stress feedback on the cellular constriction pattern.** Predictions of the AGF model for the multicellular patterns formed by constricted cells (**A**) in a system with tensile feedback ($\beta = 250$) and (**B**) in a stress insensitive random system ($\beta = 0$); in both cases 40% of cells in the active region have constricted. The tensile feedback present in (A) results in formation of chains of constricted particles, similar to cellular constriction chains observed during the initial phase of VFF.

constrictions (Fig 5B), while for neighbor-triggered constrictions, the formation of larger disconnected clumps is predicted (S1 Fig).

Similar to the CCCs observed *in vivo*, the orientation of the chain-like formations produced by our AGF model shows a preference towards extending along the anteroposterior direction. With no directional bias built into our model, this indicates that the boundary conditions of the active region play an important role in how the percolating network of constrictions is organized.

Our stress-based Voronoi tessellation model predicts that cells in constriction chains are elongated, similar to the elongated cells in *Drosophila* embryos (Fig 6). The Voronoi cells are somewhat more compact than the cells in live embryos and the CCCs are more regular; however, despite this, our simulations capture key geometrical features of the cellular constriction patterns during VFF. In contrast, the standard Voronoi construction fails to predict cell elongation (S2 Fig).

The generation and growth of constriction chains in the system with tensile feedback results from the enhanced tensile stress near the ends of already formed constriction chains [5] and from the formation of precursor tensile-stress chains that connect constriction chain fragments. We discuss these mechanisms in the following section.

## Cellular constriction chains develop along underlying precursor tensile-stress chains

We examined the time-lapse sequence of correlated apical constrictions by which CCCs emerge. Our analysis of living embryos indicates that constriction chains grow not only by adding new constricted cells to the ends of an existing chain, but also by correlated constrictions of cells separated by a larger distance. CCCs can form from an initially sparse,

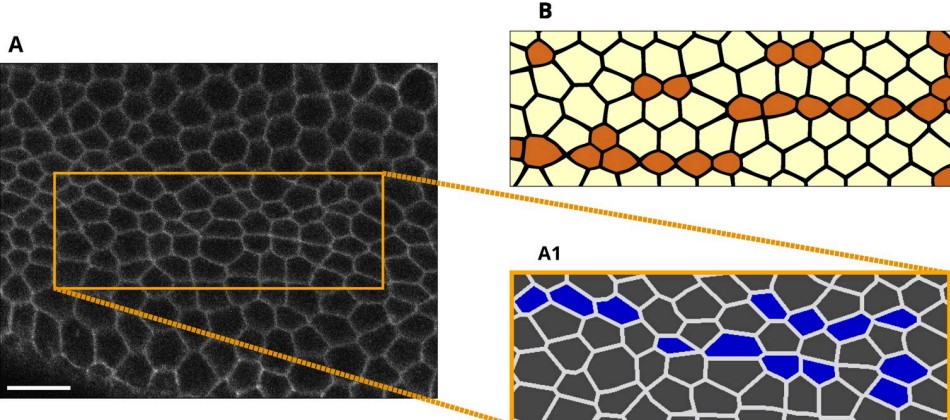

**Fig 6. Elongated shapes of cells in constriction chains.** (**A**) Image of the mesoderm primordium during VFF at time 175 s. Scale bar: 10 μm. (**A1**) Segmentation of the indicated area in (A). Constricted cells (identified using constriction cutoff $r_c = 0.65$) are highlighted in blue. (**B**) Cell shapes predicted using the AGF model with Voronoi construction using anisotropic stress-based distance function. In both the experiment and theoretical model the constricted cells are elongated and exhibit a similar constriction pattern.

disconnected group of constricted cells arranged along the contour of a constriction chain that is about to emerge (Fig 7). Unconstricted cells between such precursor chain fragments subsequently constrict, leading to the rapid generation of a connected chain-shaped cluster of constricted apices (Fig 7).

Our theoretical AGF model with tensile feedback predicts similar constriction patterns (Fig 8). An analysis of the distribution of the underlying virial stress (represented by the fill color of the cells) explains the origin of the observed constriction correlations at a distance. The overall view (Fig 8A), and the chain-formation details seen in the blowup (Fig 8B), indicate that the constriction chain formation is initiated by the presence of a precursor stress chain that involves elevated tensile stress (red) in linearly arranged unconstricted cells. Due to positive mechanical feedback, unconstricted cells along the precursor stress chain constrict, initially producing a set of isolated singlets and doublets. This strengthens the underlying precursor stress chain and eventually leads to the formation of a continuous CCC, in a process similar to the one observed *in vivo*. Most of the chains develop in this piecewise manner that results from two-way coupling between tensile stress and constriction events (Fig 8A).

The precursor stress chains are mostly arranged in the longitudinal direction because the inactive lateral regions provide only a moderate mechanical resistance. Thus, the constriction chains emerging due to tensile feedback are also mostly oriented longitudinally. This feature is consistent with the *in vivo* observations showing chains of constricted cells oriented primarily in the anteroposterior direction.

## Size distribution of constriction chains is consistent with theoretical predictions with tensile feedback

The geometric similarity between CCCs observed *in vivo* and those predicted by the AGF model suggests that mechanical feedback via tensile forces coordinates apical constrictions. To test this hypothesis, we quantitatively compared constriction patterns in the embryo to those generated by the AGF model, focusing on the number and sizes of connected clusters of constricted cells. We evaluated the normalized average number of clusters

$$C_{ave} = \langle N_{clu}/N_a^0 \rangle, \tag{12}$$

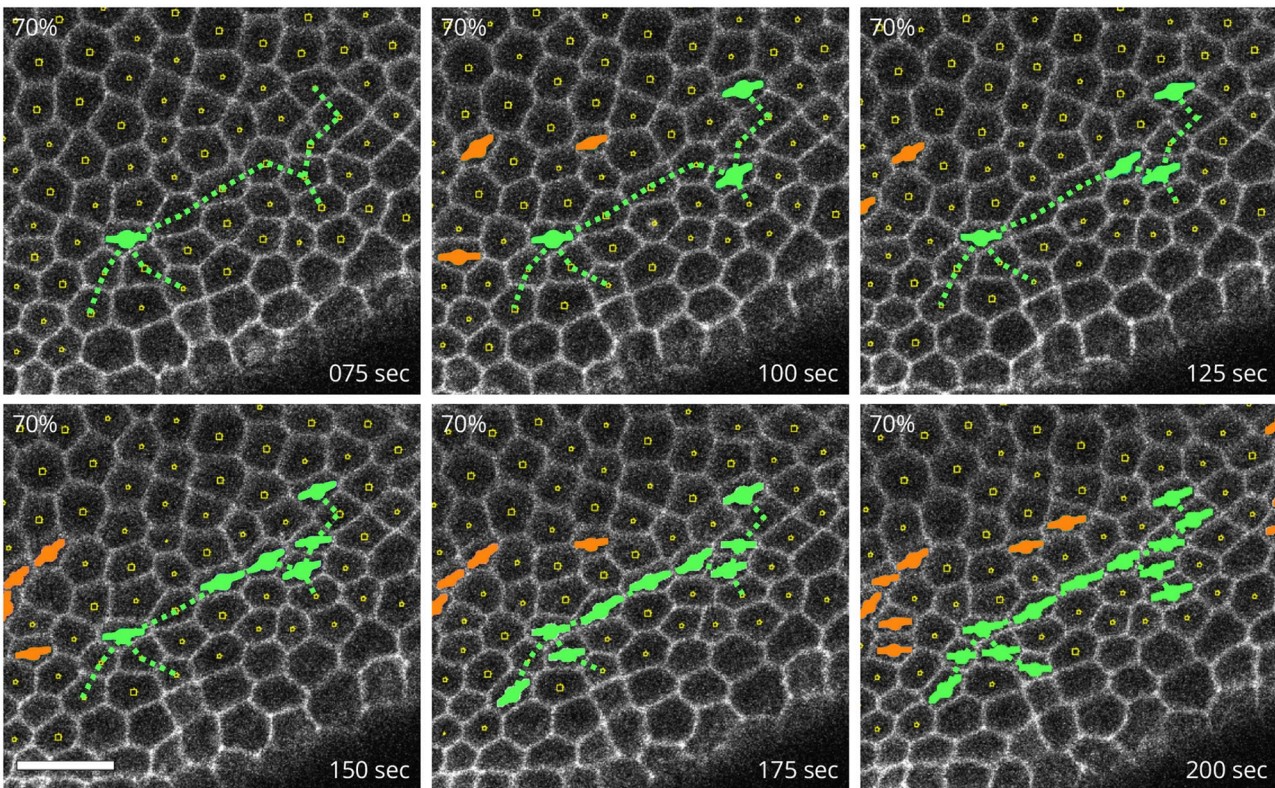

**Fig 7. Piecewise formation of a cellular constriction chain.** Processed confocal microscopy images of the ventral side of a Spider-GFP embryo during the initial slow apical constriction phase of VFF. The constricted cells highlighted in green initially form linearly arranged singlets and doublets, which are later connected into a single constriction chain. The dotted line indicates the final chain connectivity. Constricted cells are identified using the threshold $r_c = 0.70$. Scale bar: 10 μm.

and the average fraction

$$\eta_\alpha = \langle N_\alpha / N_c \rangle \tag{13}$$

of constricted cells in singlets ($\alpha$ = s), doublets ($\alpha$ = d), and multiplets ($\alpha$ = m) (where the average $\langle \cdots \rangle$ is taken over five embryos or ten simulation runs). The cluster number $N_{\text{clu}}$ is scaled by the total number of active cells $N_a^0$ (combining the already constricted and not yet constricted cells in the observed image or simulation frame), and the average number of singlets $N_s$, doublets $N_d$, and multiplets $N_m$ are normalized by the number $N_c$ of constricted cells. The normalization by the active or constricted cell number in the observed part of the system allows us to compare experimental results in which only a portion of the ventral-furrow region is imaged with simulations of the entire active domain.

We plotted $C_{\text{ave}}$ and $\eta_\alpha$ versus the fraction of active cells that have constricted %$N_c$ (Fig 9). The predictions from the AGF model are depicted for the stress-insensitive system (pink stars) and two systems with tensile-stress feedback, both having the same coupling parameter $\beta = 250$ (purple triangles), but one with strong particle constrictions (left) and the other with weak constrictions (right). The experimental data from embryos (black circles) are presented for a strong constriction threshold $r_c = 0.65$ and a weak constriction threshold $r_c = 0.85$ to shed light on the effect constriction strength plays in the constriction correlations.

Consistent with our analysis of individual simulation frames (Figs 5 and 8), the average cluster counts depicted in Fig 9 show that the tensile-stress coupling promotes the growth of

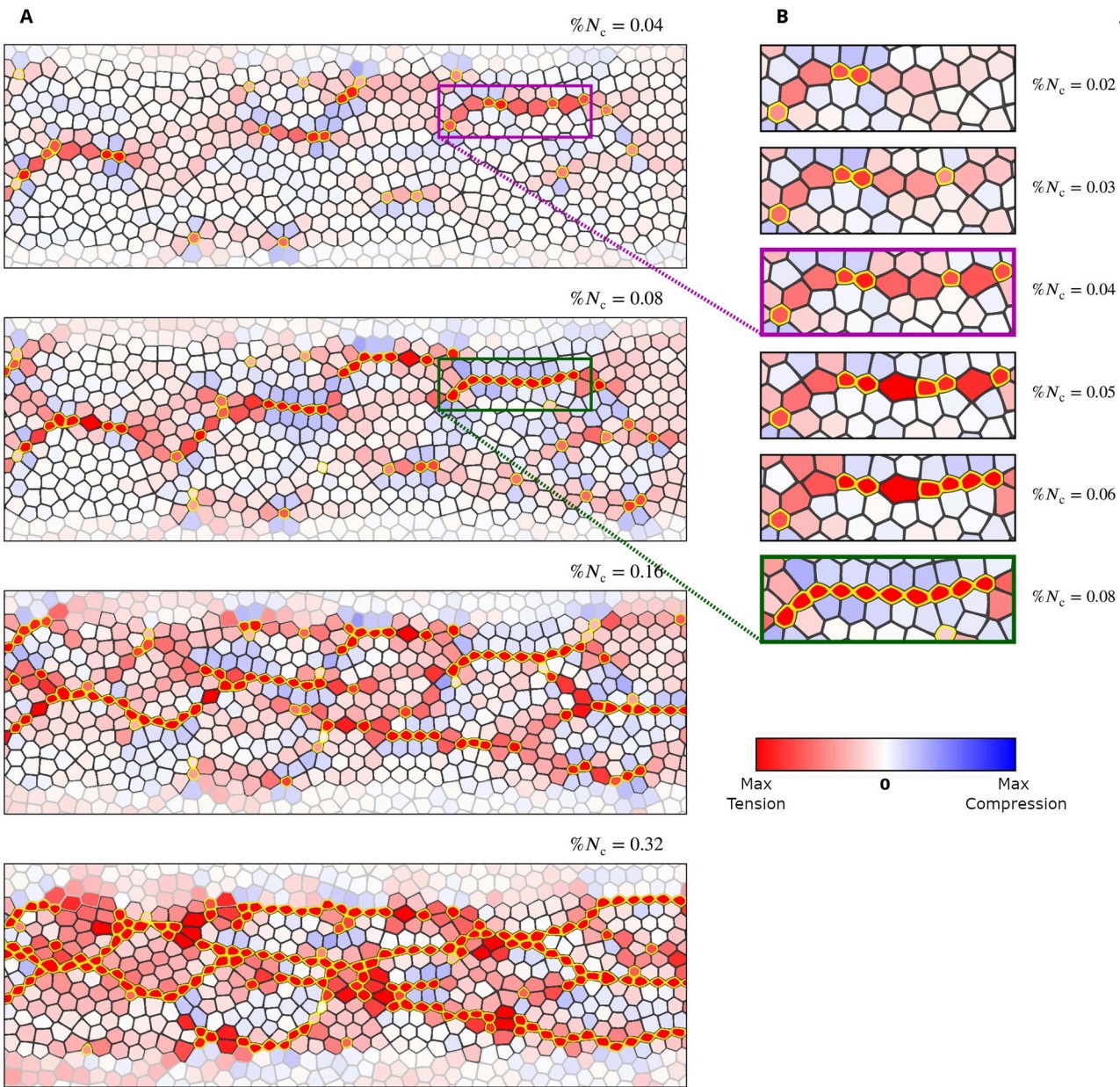

**Fig 8. Development of stress chains and constriction chains in a system with tensile-stress feedback.** Time lapse AGF simulation frames of a tensile stress sensitive system, colored to show the distribution of the major stress. The fill color of all cells shows tensile (red) and compressive (blue) major stresses, where the color saturation indicates the stress magnitude. Inactive cells (gray outline), active cells (black outline), constricted cells (yellow outline). (**A**) Overall view of the system microstructure. (**B**) Blowup of the rectangular area marked in the two top panels of (A). The simulation illustrates the two-way coupling between the stress and constrictions: cell constrictions result in formation of precursor tensile-stress chains, and precursor chains induce subsequent constrictions, leading to the formation of cellular constriction chains. System parameters: $\beta = 250$ and $f_c = 0.6$. A comparison of $\beta = 0$ and $\beta = 250$ is provided as a supplemental figure (S3 Fig).

CCCs over the creation of new clusters. In contrast, random uncorrelated constrictions produce a larger number of smaller clusters and isolated constricted cells. A similar trend occurs for strong and weak constrictions, but the effect of stress coupling is much more pronounced for strong constrictions (i.e., the weaker constriction plots show less difference between the tensile feedback and random curves than is observed for the strong constriction case). This is

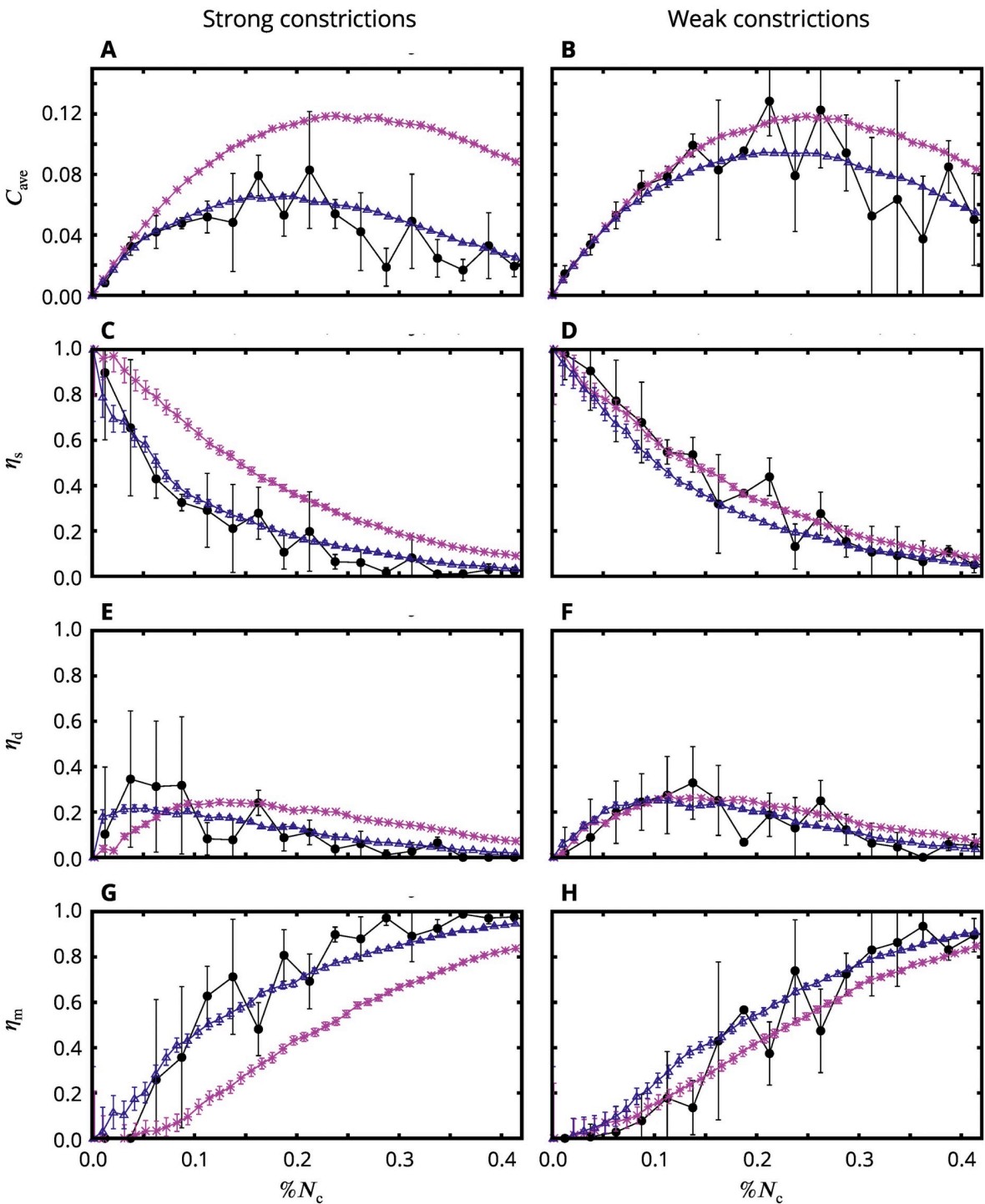

**Fig 9. Constricted-cell cluster counts in the AGF model and *in vivo*.** (**A,B**) Normalized average number of clusters $C_{ave}$, (**C,D**) average fraction of particles in singlets, $\eta_s$, (**E,F**) doublets, $\eta_d$, and (**G,H**) multiplets, $\eta_m$. Simulations with strong constrictions (left) weak constrictions (right) vs the fraction of active constricted cells $\%N_c$. Tensile feedback (purple triangles); random uncorrelated system (pink stars). The stress sensitive system uses the coupling parameter $\beta = 250$ and constriction factors $f_c = 0.6$ (strong constrictions) and $f_c = 0.85$ (weak constrictions). Strong constrictions *in vivo* were identified using the constriction threshold $r_c = 0.65$, and weak constrictions using $r_c = 0.85$. The experimental data (black circles) agree with the numerical predictions for the system with tensile-stress feedback but not with the random uncorrelated system. Embryos ($n = 5$); simulations ($n = 10$). Error bars: SD. For simulations, the error bars are smaller than the symbols representing the data points.

not unexpected as weaker constrictions will inherently introduce lower stress per constriction and ergo less tensile-stress feedback.

The degree by which the number of clusters is reduced in the system with tensile feedback relative to the random system increases with the progress of the constriction process (Fig 9A and 9B). This significant reduction (for strong constrictions up to about 75% at %$N_c$ = 0.4) reflects gradual formation of a percolating network of constriction chains via connection of singlets and smaller clusters along the underlying tensile stress lines (Fig 8). Correspondingly, the number of singlets is significantly reduced (Fig 9C and 9D), and the number of constricted cells in multiplets is increased (Fig 9G and 9H). Both in the random and stress-correlated systems the number of doublets initially increases and then decreases as the doublets are removed from the population by growth into larger clusters due to constrictions of adjacent neighbors (Fig 9E and 9F); this behavior, however, is significantly more pronounced in the presence of stress feedback.

We observe that experimental data for constricted cell clusters in live embryos agree, within the statistical inaccuracies, with the theoretical predictions for the system with tensile feedback but not with those for the uncorrelated system with no feedback. This clear trend is consistently seen for all quantities considered, i.e., for the normalized number of clusters and the distribution of singlets, doublets, and multiplets. In the strong-constriction case, the experimental data differ from the results for the uncorrelated random system considerably beyond the standard deviation.

The essential qualitative features of the cluster distribution predicted theoretically for the tensile-stress correlated constrictions are clearly present in the experimental curves. In particular, for strong constrictions the fraction of singlets quickly decays when the constriction process progresses (Fig 9C); the initial rapid growth of the population of doublets is followed by a decrease to nearly zero (Fig 9E); and a large fraction of constricted cells form multiplets (Fig 9A and 9G). Furthermore, a good agreement is also observed for weaker constrictions: As predicted, the experimental singlet and doublet counts (Fig 9D and 9F) are close to the random distribution. The average cluster size (Fig 9B) initially follows the random distribution and at the later stage of apical constrictions falls below the random case, consistent with the theoretically predicted delay of formation of the constricted-chain network.

Taken together, the results obtained using strong and weak constriction thresholds present a cohesive picture of the initial slow phase of apical constrictions. The weak constriction thresholds show us the beginning where cells pulse stochastically, generating fleeting bursts of tensile stress through the mesoderm primordium. These relatively weak stress bursts combine constructively to trigger some cells to ratchet their constrictions. The strong constriction threshold then describes cells performing ratcheted apical constrictions that generate a consistent underlying stress field that permeates the mesoderm primordium, promoting the triggering of ratcheted apical constrictions along paths of high tensile stresses. Since many subtle features of the observed apical-constriction patterns are predicted by our tensile feedback model, our results not only show that apical constrictions are spatially correlated but also provide evidence that these correlations are a consequence of mechanical feedback via tensile stress.

## The AGF model predicts that tensile feedback aids robustness of the cellular constriction process

Previous sections established that apical constrictions during the slow phase of VFF are coordinated by mechanical feedback and described details of the emerging cellular constriction chain patterns. Here we ask whether the coupling between apical constrictions and the tensile stress

field may aid robustness of VFF. In live embryos contractility fluctuations occur naturally due to randomness of biophysical processes or from embryonic defects. Therefore, ensuring consistent and robust outcomes of the slow phase of VFF, which prepares the mesoderm primordium for subsequent invagination, requires a control mechanism to coordinate cellular constrictions across the tissue and synchronize constriction levels between different regions. Based on our theoretical results presented below, we show that tensile feedback can provide such a control mechanism and reduce constriction nonuniformities in the presence of spatial variation of cell contractility.

To theoretically predict possible effects of tensile feedback on robustness of VFF, we consider a model system with locally reduced magnitude of the constriction probability function. The reduction is achieved by decreasing the probability amplitude $\alpha_i$ in Eq 8 for active cells in a prescribed disrupted region,

$$\alpha_i = \begin{cases} 1 & \text{cells outside the disrupted region,} \\ \alpha_r < 1 & \text{cells in the disrupted region.} \end{cases} \tag{14}$$

In our simulations $\alpha_r$ ranges from zero (complete contractility disruption) to 0.6 (moderate disruption). We present results for a band of affected active cells (Figs 10 and 11) and a region in the shape of an ellipse (Fig 12). The depicted constriction patterns and stress distributions demonstrate that tensile-stress sensitive systems show constriction recovery via formation of CCCs whereas stress insensitive systems do not recover.

For systems with a disrupted band of cells (Figs 10 and 11), constriction patterns and stress distribution outside the disrupted domain are not significantly affected by the presence of the disrupted zone. We thus focus on the disrupted region (between the two vertical lines) where the probability amplitude has been reduced. In the purely random case the fraction of constricted cells in the affected band is decreased by a factor that is commensurate with the imposed reduction in the constriction probability (Fig 10B). However, cellular constrictions in the disrupted band are much less disturbed if the tensile feedback is present (Fig 10A). For example, in the uncorrelated random system with $\%N_c = 0.37$ and $\alpha_r = 0.2$ (constriction probability reduced by 80%), there are only a few constricted particles in the affected domain, but there are two constriction chains crossing the affected zone in the system with stress feedback. A similar behavior is observed for weaker probability amplitude reduction. In particular, in the tensile-stress sensitive system with a moderate probability reduction, $\alpha_r = 0.6$, the only distinct feature of the disrupted region is some additional buildup of tensile stresses there; otherwise the affected region is indistinguishable from the surrounding unaffected regions.

To quantitatively compare the constriction levels in the affected band of cells and the unaffected region we define the constriction perturbation coefficient

$$\chi_{cp} = \frac{\%N_c^t}{\%N_c^u}, \tag{15}$$

where $\%N_c^t$ denotes the fraction of active cells that constricted in the tested domain (the affected band in the current case) and $\%N_c^u$ is the fraction of active cells that constricted in the unaffected region. According to the above definition, $\chi_{cp} = 1$ indicates that the disrupted zone recovers to the normal behavior.

At the beginning of the constriction process $\chi_{cp} \approx \alpha_r$ for both random and stress-sensitive systems; the constriction perturbation coefficient increases with the progress of constrictions, but this behavior is very different for the random and stress-correlated systems (Fig 10C). Without tensile feedback the increase results from the gradual saturation of constrictions in

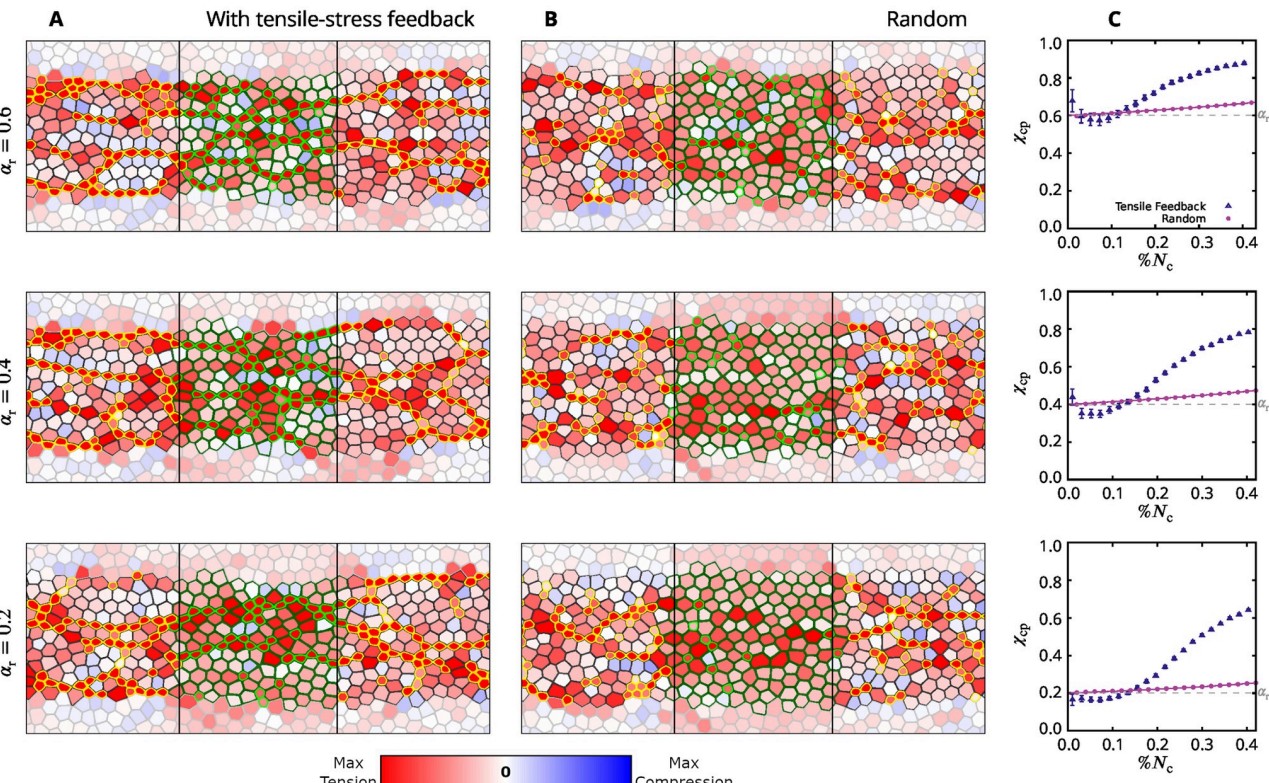

**Fig 10. The effect of tensile feedback on constrictions in a disrupted zone with decreased constriction probability. (A,B)** Simulation snapshots for the AGF model with a band of cells whose constriction probability is decreased by the factor $\alpha_r = 0.6$ (top) 0.4 (middle) and 0.2 (bottom) relative to the probability outside the affected zone. The results are shown for the system with tensile feedback (A) and the stress insensitive random system (B). The affected active cells with reduced constriction probability are outlined in green; otherwise the images follow the color convention used in Fig 8. Frames correspond to $\%N_c = 0.37$. The extent of the disrupted zone is indicated by two vertical black lines. **(C)** Plots showing the constriction perturbation coefficient $\chi_{cp}$, Eq 15, vs the fraction of constricted cells $\%N_c$ for a given value of $\alpha_r$. Simulation results (symbols); exact theoretical result, Eq 18, for uncorrelated random constrictions (solid lines). The thin dashed lines represent the values of $\alpha_r$. The coefficient $\chi_{cp}$ reflects how well the affected region is performing compared to the unaffected areas, with $\chi_{cp} = 1$ indicating full recovery of the affected region. All three plots show that the system with tensile feedback exhibits substantial constriction recovery at sufficiently large values of $\%N_c$. Tensile-sensitive simulations ($n = 120$); random simulations ($n = 16000$). Error bars: SE.

the unaffected region (as described by Eq 18) and is slight for $\%N_c$ below 40%. In the presence of tensile feedback, however, $\chi_{cp}$ increases rapidly after an initial delay, and is considerably higher than the one for random constrictions. Consider, for example, the data points that correspond to the fast-phase threshold $\%N_c = 0.4$. For a moderate value of the probability reduction, $\alpha_r = 0.6$, the simulations of the system with tensile feedback yield $\chi_{cp} \approx 0.9$, i.e., there is nearly complete constriction recovery at the development stage corresponding to the onset of the fast VFF phase *in vivo*. For a system with $\alpha_r = 0.4$ we have $\chi_{cp} \approx 0.8$. Even for the strongest disruption, $\alpha_r = 0.2$, we obtain $\chi_{cp} \approx 0.6$, which shows that with the help of tensile feedback three times more cells constricted in the strongly disrupted region than they would without the feedback.

Constrictions, which initially occur with higher frequency in the unaffected domain, cause gradual buildup of tensile-stress chains in the affected region (Fig 11). Thus, due to the mechanical feedback, the constriction probability is elevated for the affected cells under tension. Hence, CCCs penetrate the affected zone along the tensile-stress chains, rescuing the constriction process (Fig 11). For strong disruption the onset of recovery is delayed because more intense stresses are needed for the reduced constriction probability to be sufficiently elevated

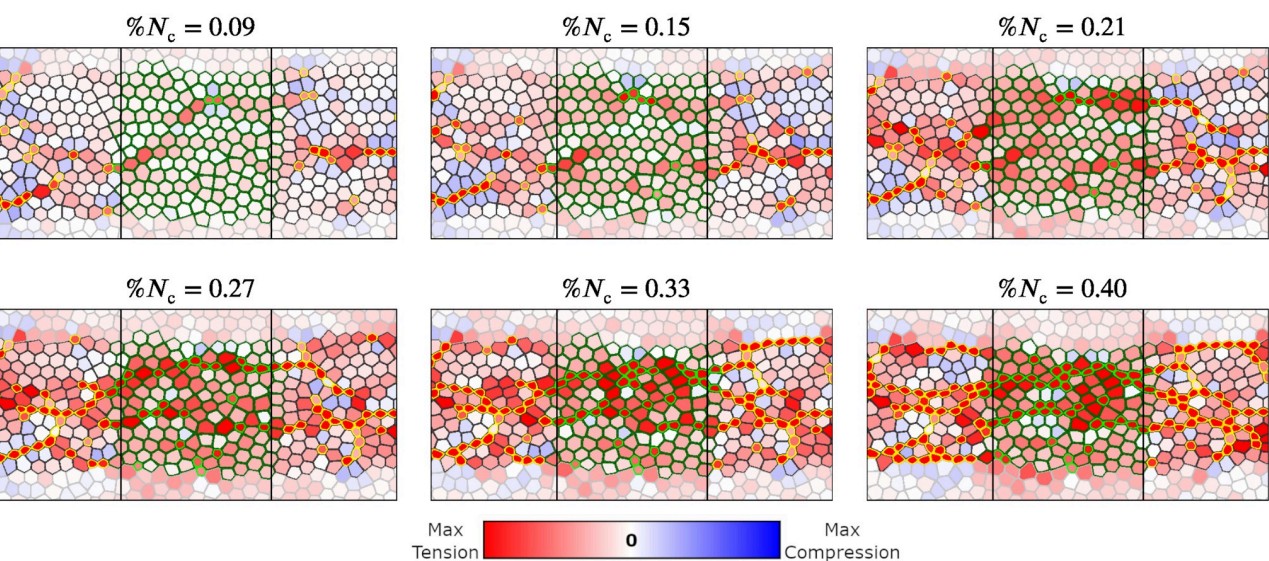

**Fig 11. The mechanism of constriction chain penetration into the disrupted zone in a system with tensile feedback.** Time-lapse images for the AGF system with a band of cells whose constriction probability is reduced by 80% ($\alpha_r = 0.2$). Color convention per Fig 10. Top frames show the initial formation of precursor tensile-stress chains crossing the affected region. Due to tensile feedback the constriction probability is elevated along these precursor chains. Thus cellular constrictions follow, resulting in formation of constriction chains crossing the disrupted zone (bottom frames).

due to feedback. These findings are consistent with the results of laser cutting experiments that isolate a region in the mesoderm primordium similar to our simulated band [16].

Here, we address the progression of apical constrictions in a tensile-stress sensitive system with an ellipsoidal affected region at the center of the active domain (Fig 12). The lateral size of the ellipse is about 75% of the width of the active region, so there are unaffected cells above and below the ellipse. The time-lapse images are presented for $\alpha_r = 0.4$ (a partial probability reduction) and $\alpha_r = 0.0$ (the complete reduction of the constriction probability). The depicted simulation frames indicate that constriction chains promote recovery of a balanced constriction pattern in two ways, depending on how strongly the affected region is perturbed. For the complete constriction probability reduction, $\alpha_r = 0$, the affected cells cannot constrict. In this case we observe that constriction chains wrap around the affected ellipse. In spite of the severity of the local constriction disruption, formation of the envelope of constriction chains around the affected region results in a relatively uniform constriction pattern. For $\alpha_r = 0.4$ constriction recovery occurs by a combination of the wrap-around mechanism and the penetration of constriction chains into the affected region after the stress chains build up.

To quantitatively characterize the chain-penetration and wrapping-around mechanisms by which tensile-stress feedback rescues a balanced constriction pattern, we consider three complementary measures of recovery. The first measure (denoted $\chi_{cp}^{E}$) is the constriction perturbation coefficient $\chi_{cp}$ evaluated for the affected elliptical region. This quantity characterizes constriction rescue by the penetration of CCCs into the affected region. The second measure (denoted $\chi_{cp}^{B}$) is the constriction perturbation coefficient $\chi_{cp}$ evaluated for the unaffected bypass regions above and below the affected ellipse (in the area between the two vertical lines in Fig 12A and 12B). This measure shows the effect of the disrupted ellipse on constriction in the surrounding unaffected domains ($\chi_{cp}^{B} > 1$ corresponds to constriction enhancement). The third measure (denoted $\chi_{cp}^{EB}$) is the constriction perturbation coefficient in the entire band of

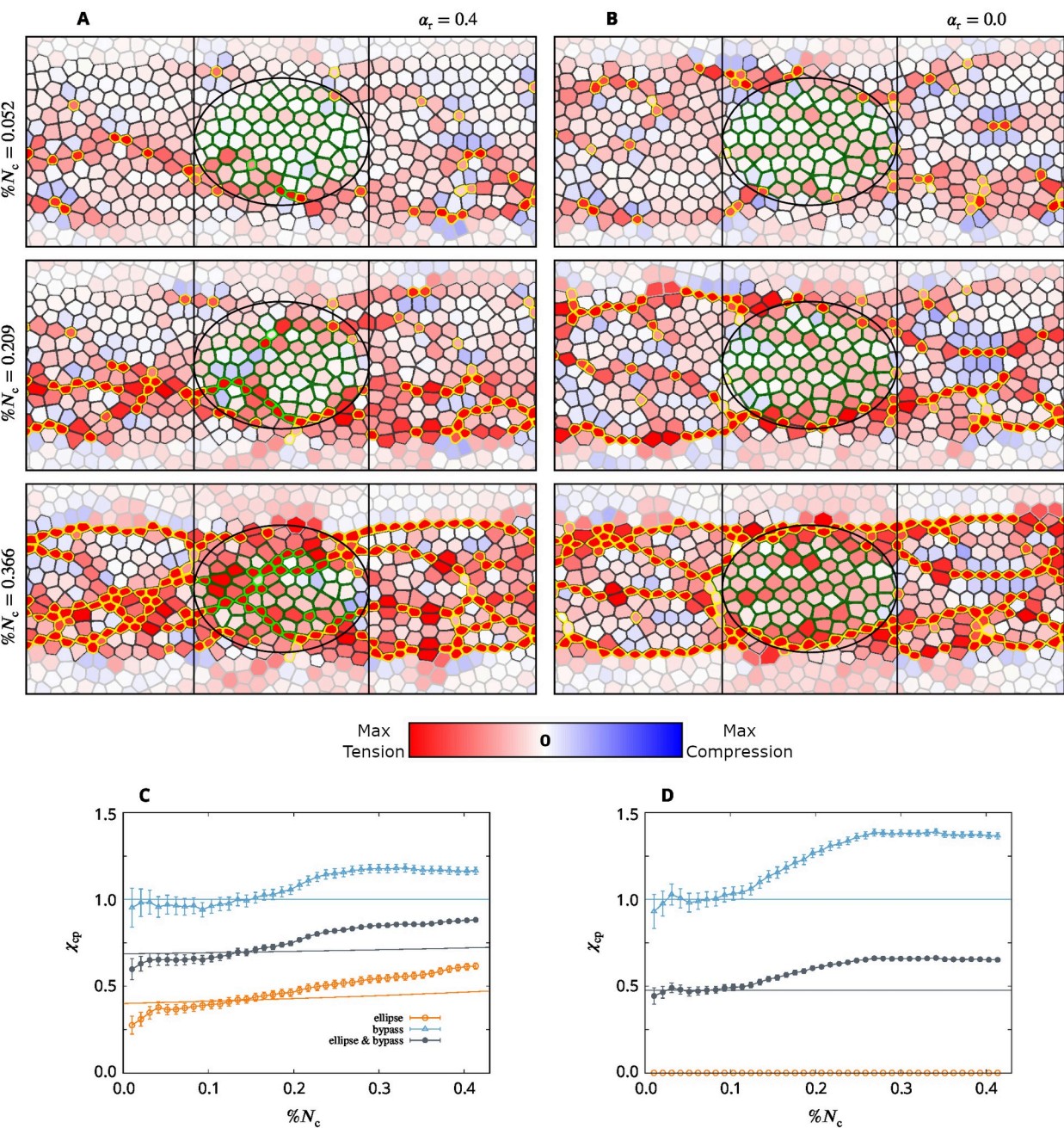

**Fig 12. Two modes of constriction rescue for elliptical disrupted domain in a system with tensile feedback.** (**A,B**) Time-lapse simulation frames for the AGF system with an elliptic region of cells whose constriction probability is (A) decreased by a factor $\alpha_r = 0.4$ and (B) reduced to zero, $\alpha_r = 0$. Color convention per Fig 10. The disrupted area is indicated by the ellipse. (**C,D**) Corresponding measures of the constriction recovery for the ellipse $\chi_{cp} = \chi_{cp}^E$ (orange), bypass region $\chi_{cp} = \chi_{cp}^B$ (blue), and joint ellipse and bypass region $\chi_{cp} = \chi_{cp}^{EB}$ (the entire area between the vertical lines; slate). Simulation results for the tensile sensitive system (symbols); exact theoretical results, Eq 18 and Eq 19, for uncorrelated random constrictions (lines). The results show that for the complete probability reduction the constriction recovery in the tensile sensitive system occurs by constriction chains wrapping around the affected ellipse, and for the partial reduction constriction chains both penetrate and wrap around the affected region. Simulations ($n = 120$). Error bars: SE.

active cells that encompasses both the affected region and the bypass regions. It includes contributions from constriction chains that penetrate or wrap around the affected ellipse.

Plots of the constriction perturbation coefficients vs $\%N_c$ for the constriction probability reduction $\alpha_r = 0.4$ and $0.0$ imply strong recovery of cellular constrictions in the affected region before the system reaches the fast-phase threshold $\%N_c = 0.4$ (Fig 12C and 12D). For the moderate probability reduction $\alpha_r = 0.4$ the combined CCC penetration and bypass contributions yield $\chi_{cp}^{EB} \approx 0.9$, indicating nearly complete constriction recovery. In the case of the probability reduction to zero there are no constrictions in the affected region, but the parameter $\chi_{cp}^{B}$ shows that there is a significant recovery of the constriction pattern by the wrap-around mechanism.

The above analysis demonstrates that tensile-stress feedback may offer robustness of VFF by providing a mechanism to even out irregularities of the pattern of constrictions. Uneven constriction distribution would result in corresponding fluctuations of the timing and degree of invagination along the active region, causing formation of an irregular or defective furrow. The rescue mechanism does not necessarily have to provide a solo means of full system recovery of a strongly disrupted system to be beneficial. Similar to how chemical buffers only need maintain pH for small amounts of an added acid to be useful, tensile-stress feedback recovery of single cells or small groups of cells for moderate contractility fluctuations aids robustness of the invagination process.

Since local variation of cell contractility *in vivo* was produced by Guglielmi *et al.* [25] using optogenetic techniques, we now discuss their results in the context of our theoretical predictions. We show below that the results of our calculations for the tensile-stress sensitive system with locally lowered constriction probability are strikingly similar to the corresponding experimental findings.

## Optogenetic experiments provide evidence of robustness of apical constrictions in live embryos

To locally affect the ability of cells to constrict during VFF in the *Drosophila* embryo Guglielmi *et al.* [25] developed an optogenetic approach targeting PI(4,5)P2 to rapidly deplete actin from the embryonic cell cortex to reduce actomyosin contractility in a photoactivated region. Illuminating the embryo using different laser powers allowed them to vary the degree of the local cell contractility inhibition, thus affecting the morphology of the developing tissue to a varying degree. Guglielmi *et al.* [25] used three different levels of laser power to optogenetically modulate the contractility in a band of cells in the central part of mesoderm primordium (Fig 13 [25]). The photoactivated band is analogous to the affected zone in our AGF model. Applying the lowest power level of 0.7 mW did not prevent the ventral-furrow invagination, although there is a clear defect in the furrow morphology (Fig 13I–13L). For higher levels of the laser power, 1.5 mW (Fig 13E–13H) and 3.0 mW (Fig 13A–13D), there was no transition to the fast phase of VFF and the embryo failed to invaginate.

The lowest level of the optogenetic activation (0.7 mW laser power) reveals constriction chains crossing the affected region of reduced cell contractility in the middle part of the frame (Fig 13J and 13K, and S5 Fig). For wild-type control, please see S6 Fig. While the contrast of the images is insufficient for a quantitative comparison, the fraction of constricted cells in the unaffected and optogenetically affected regions appears to be approximately the same, consistent with our theoretical predictions for the rescue of the constriction process in the system with tensile feedback.

A similar conclusion can be drawn from an analysis of the images of the embryo illuminated with a moderate power 1.5 mW laser beam. In addition to a visual inspection, we performed a quantitative analysis of experimental data in this case, using three high-contrast

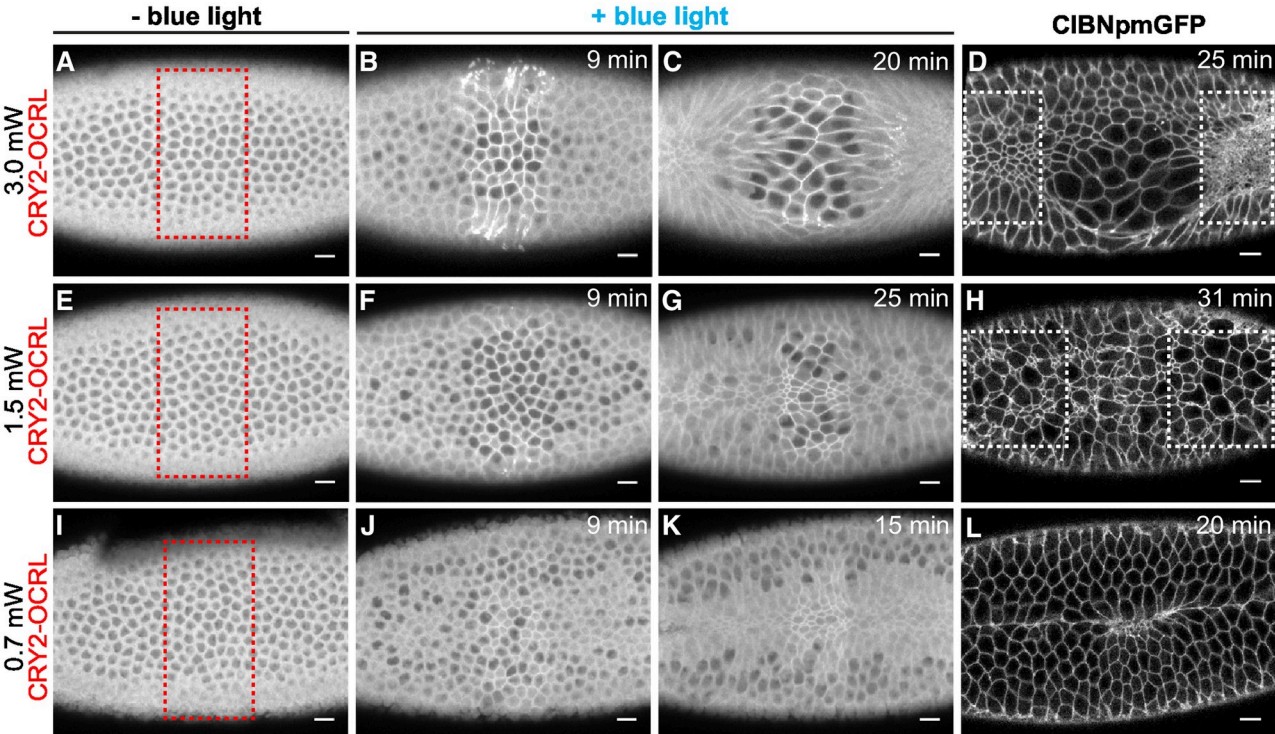

**Fig 13. Confocal images of representative embryos with a zone of optogenetically reduced contractility.** The apical surface of the ventral mesoderm is shown during the apical-constriction process. To locally reduce cell contractility, the embryos were exposed to blue light in the zone indicated by the red box in left panels. The affected zone was photoactivated with a (**A-D**) 3 mW, (**E-H**) 1.5 mW, and (**I-L**) 0.7 mW laser beam; the degree of cell contractility reduction is commensurate with the beam power. The first three images in each row are video frames from a confocal movie, and the last frame is a high-resolution confocal image. Time after photoactivation as indicated. For 0.7 mW and 1.5 mW laser power, the images show a significant number of constricted cells in the affected region, forming chain-like arrangements (E-L). Modified versions of (F-G) and (J-K) with constricted cells manually identified are provided as supplemental figures (S4 and S5 Figs). For the 3.0 mW laser power, constricted cells wrap around a large cluster of unconstricted (expanded) cells (A-D). Scale bars: 10 μm. Figure reprinted from [25] under the article's CC BY license.

images (courtesy of Guglielmi *et al.* [25]) available for the final state of the constriction process. A comparison of images of weak and moderate photoactivation at the 9 minute mark (Fig 13F and 13J, S4 and S5 Figs) shows delayed apical constriction in moderate reduction of contractility. However, images at 25 min. (Fig 13G and S4 Fig) show that a significant number of constricted cells are present in the disrupted zone, and at 31 min. (Fig 13H) there are even some hyperconstricted cells. It follows that cells in the affected region can still constrict strongly, but such constrictions occur with a delay. This behavior is similar to our simulations, which demonstrate that in more strongly affected systems more time is needed to build sufficient tensile stress to rescue constrictions.

After a sufficiently long time, the fraction of constricted cells in the affected zone is approximately the same as the corresponding fraction outside the affected region (Figs 13H and 14A). Thus the constriction process has been rescued, in spite of significant disruption of cell contractility. This conclusion is supported by a quantitative analysis of three embryos, which do not show any deficit of constricted cells in the affected domain, in spite of a significant disruption of the ability of cells to constrict (Fig 14B).

The relatively consistent constriction pattern across 1.5 mW embryos (with affected and unaffected regions behaving similarly despite a significant disruption of myosin contractility) was recognized by Guglielmi *et al.* [25] and attributed to the fact that some cells within the photoactivated area retained the capability to contract while others did not. This behavior can,

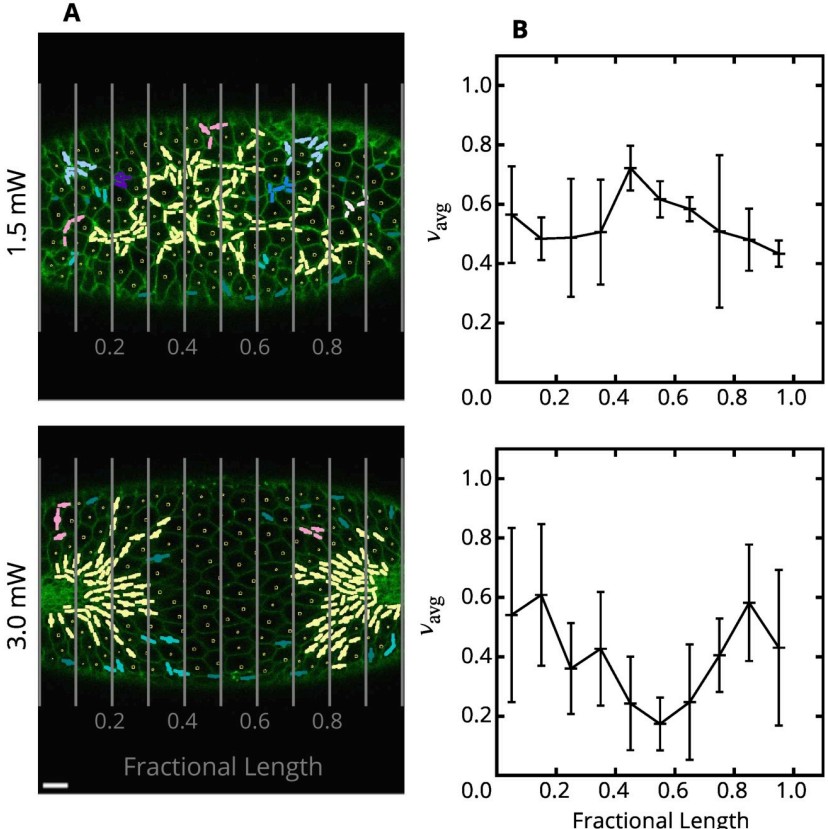

**Fig 14. Fractions of constricted cells in optogenetically disrupted embryos.** (**A**) Representative images of embryos with a band of cells about their length center optogenetically activated with 1.5 mW (top) and 3.0 mW (bottom) laser (raw data from Guglielmi *et al.* [25]). Cells are considered constricted if their minor axis length is 1.3 μm or less. The colors represent the number of cells belonging to a single constricted-cell cluster according to the scale defined in Fig 2. (**B**) Averaged binned fraction of constricted cells, $v_{avg}$ (normalized by the total number of cells in the analyzed frames), vs fractional position along the observed portion of the mesoderm primordium, as defined by the vertical lines in the corresponding images shown in (A). Due to strong photoactivation the analyzed embryos did not undergo the transition to the fast constriction phase and failed to invaginate; the images were taken at a time after invagination would normally occur. Photoactivation by a 1.5 mW laser beam does not show a constriction deficit in the affected region, indicating a recovery of contractile activity consistent with our predictions. Photoactivation by 3.0 mW shows a significant deficit of constrictions in the affected zone, but the contractile activity does not drop to zero because fragmented constriction chains tend to wrap around the disrupted region. Scale bar: 10 μm. Embryos ($n = 3$) for each experiment. Error bars: SD.

however, be explained in terms of mechanical feedback. The optogenetically affected cells in the 1.5 mW embryos are disrupted to a degree that the formation of the underlying stress field is impaired, but not fully suppressed. Ratcheted constrictions are still triggered along the underlying precursor tensile-stress chains, but cells must constrict more than usual to support enough stress buildup to trigger constrictions of photoactivated cells. Visually, this produces a combination of hyperconstricted and enlarged cells in the unaffected areas. Once chains penetrate the affected area, a similar process will begin to promote percolation. Our simulations (Fig 10) predict a substantial recovery of constrictions even at 80% reduction of the constriction probability when the constrictions are correlated via tensile feedback.

We introduce the fraction of constricted cells, $v_{avg}$, which is similar in spirit to %$N_c$ but differs in that it relies on information from a single frame (or subregion of a single frame) rather than across a timelapse. We define $v_{avg}$ as the fraction of constricted cells over the total number

of cells while %$N_c$ is the fraction of active cells that constricted. In the analyzed frames, $v_{avg}$ is larger than 40% (Fig 14B) because the transition to the second fast phase of VFF has not occurred, and apical constrictions continued beyond the usual 40% slow-to-fast-phase transition threshold. Progression of the slow phase of constrictions beyond the time at which invagination usually takes place is likely to be responsible for strong bidispersity of cell sizes (Fig 13H). Specifically, the cells in already formed constriction chains continued to constrict, causing the others to expand. A similar behavior can be seen in images of embryos with delayed or failed VFF due to injection of Rho kinase inhibitor [42].

The 3.0 mW laser beam hinders constrictions in the affected zone almost entirely (Fig 13A–13D). The optogenetic disruption leaves the cells unable to perform active mechanical responses, and as a result, they stretch and enlarge when being pulled. However, as already noted by Guglielmi *et al.* [25], the cells bordering the affected region still constrict; this behavior is similar to the constriction pattern seen in our simulations for the ellipsoidal affected domain, where tensile-stress sensitivity results in formation of constriction chains wrapping around the region of hindered contraction probability (Fig 12B). Even at 3.0 mW illumination there is some constriction recovery due to the wrap-around effect (Fig 14B).

In summary, the results of the optogenetic experiments by Guglielmi *et al.* [25] agree well with predictions from our theoretical analysis of the role of mechanical feedback in coordinating apical constrictions to enhance robustness of VFF. In particular, the experiments show that chains of constricted cells either penetrate or wrap around the affected region, depending on the degree of hindrance of cell contractility. Therefore, there is a substantial recovery of constrictions in the optogenetically affected band of cells. These phenomena, predicted by our AGF model, would be difficult to explain without the assumption that the tensile-stress feedback coordinates cell constrictions.

## Discussion

### Tensile stress chains coordinate apical constrictions in VFF

The formation of the Drosophila ventral furrow (VF) is initiated by flattening at cell apices [17, 22] in a field of cells specified by the dorsoventral patterning system to be the mesoderm primordium [43]. In the next stage of this multi-step morphological change, the apical actomyosin cytoskeleton undergoes contractile pulses of three types: unconstricting, unratcheted constricting, and ratcheted constricting pulses [21, 24]. Ratcheted pulses result in progressive apical constriction of cells [24] in which they occur. Originally thought to be random in nature [22], it was later found that these constrictions are correlated, and that neighbors of cells that have undergone ratcheted apical constrictions are more likely to constrict than non-neighboring cells [24]. The investigations described here and in our recent work [5] show that correlations between apical constrictions occur at a higher order of organization than constrictions of neighboring cells. Our theoretical analysis indicates that isotropic neighbor-to-neighbor correlations would produce grainy patterns of constricted cells (see S1 Fig). However, we observe that apical constrictions tend to organize into chain structures that form roughly linear patterns oriented along the anterior-posterior axis before the transition to the fast phase of apical constrictions, initiated by the Fog and T48 signaling pathways.

Based on a theoretical analysis of constriction patterns with and without stress coupling, we conjectured [5] that anterior-posterior-biased chaining results from tensile-stress feedback that coordinates apical constrictions. The results presented here provide strong support for this conclusion. Cellular constriction chains form along a scaffolding of underlying paths of aligned tensile stress and predominantly grow along the anterior-posterior axis due to

anisotropic mechanical stresses. Tensile stress along the anterior-posterior axis of the VF primordium was shown by laser cutting experiments [33], consistent with our modeling efforts.

Our observations of formation of constriction chains *in vivo* and our numerical simulations based on the tensile feedback assumption show that development of CCCs often occurs through initial formation of disconnected smaller clusters that are later connected into a single larger chain. We predict that these smaller clusters lie along the path of aligned precursor tensile stress and form as a result of mechanical feedback that coordinates constrictions over distances that span multiple cells. Cellular constrictions along the precursor tensile-stress chain cause a further stress-chain enhancement (Fig 15). Thus, there is a two-way coupling between constriction events and the associated stresses, leading to the rapid growth of a percolating stress-bearing constriction network.

A detailed comparison of the measured distribution of clusters of constricted cells *in vivo* with theoretical predictions for a system with tensile stress sensitivity showed a quantitative agreement between the theory and the experiments for the entire duration of the slow phase of VFF (Fig 9). This is seen for both strong and weak constrictions; these two cases have different statistics due to a different magnitude of tensile stresses produced by the constricted cells. Moreover, analysis of Guglielmi *et al.*'s [25] optogenetic experiments shows that the theoretically predicted recovery of apical constrictions in a zone of reduced contractility indeed occurs *in vivo*, further supporting our tensile feedback conjecture.

Our theoretical studies show that at the end of the slow apical constriction phase, tensile stress through the mesoderm primordium is primarily carried by the network of CCCs. Moreover, the regions of unconstricted cells between the chains generally form areas of low stress (Fig 8). We expect that the reduced cellular stress is conducive for the later signaling-mediated rapid constriction of the VF placode immediately preceding invagination. It is possible that the unconstricted cells in the low stress pockets are able to constrict with less resistance and at a lower energy cost than in an uncorrelated system. In the absence of mechanical feedback, many unconstricted cells would be subjected to a strong tensile stress (S3 Fig), because there is no percolating constricted-cell network to carry the load.

It is likely that the stress carried along the network of CCCs provides a useful mechanical coupling between different sections of the forming furrow. We anticipate that carrying tension along the anterior-posterior axis plays an important role in coordinating and navigating the formation of a uniform invagination. Additionally, the distribution of tensile forces helps to prevent any wrinkles or puckering as the mesoderm primordium buckles inwards. To verify this physical picture, we are working on a particle-based 3D invagination model.

Optogenetic experiments offer an elegant and tunable perturbation that can simulate a variety of situations. Mechanical perturbations, stochastic biochemical or genetic abnormalities, and environmental perturbations such as temperature or toxins can all be mimicked. Optogenetic approaches can also address mutations affecting dorsoventral patterning, mesoderm primordium specification (e.g. *sna*, *twi*), actomyosin components and regulators, and adherens junction components. Our analysis of Guglielmi *et al.*'s optogenetic experiments [25] indicates that the tensile-feedback coupling between different sections of the VF mesoderm primordium is especially important in systems with regions of reduced actomyosin contractility. We observe that constriction chains can propagate across regions of reduced actomyosin contractility, in accordance with the predictions of our model. The experimental data show that for moderate photoactivation a significant constriction recovery occurs by penetration of cellular constriction chains into the affected zone; whereas for strong photoactivation constriction chains are redirected to circumvent the region of impaired contractility. Both phenomena agree with our model.

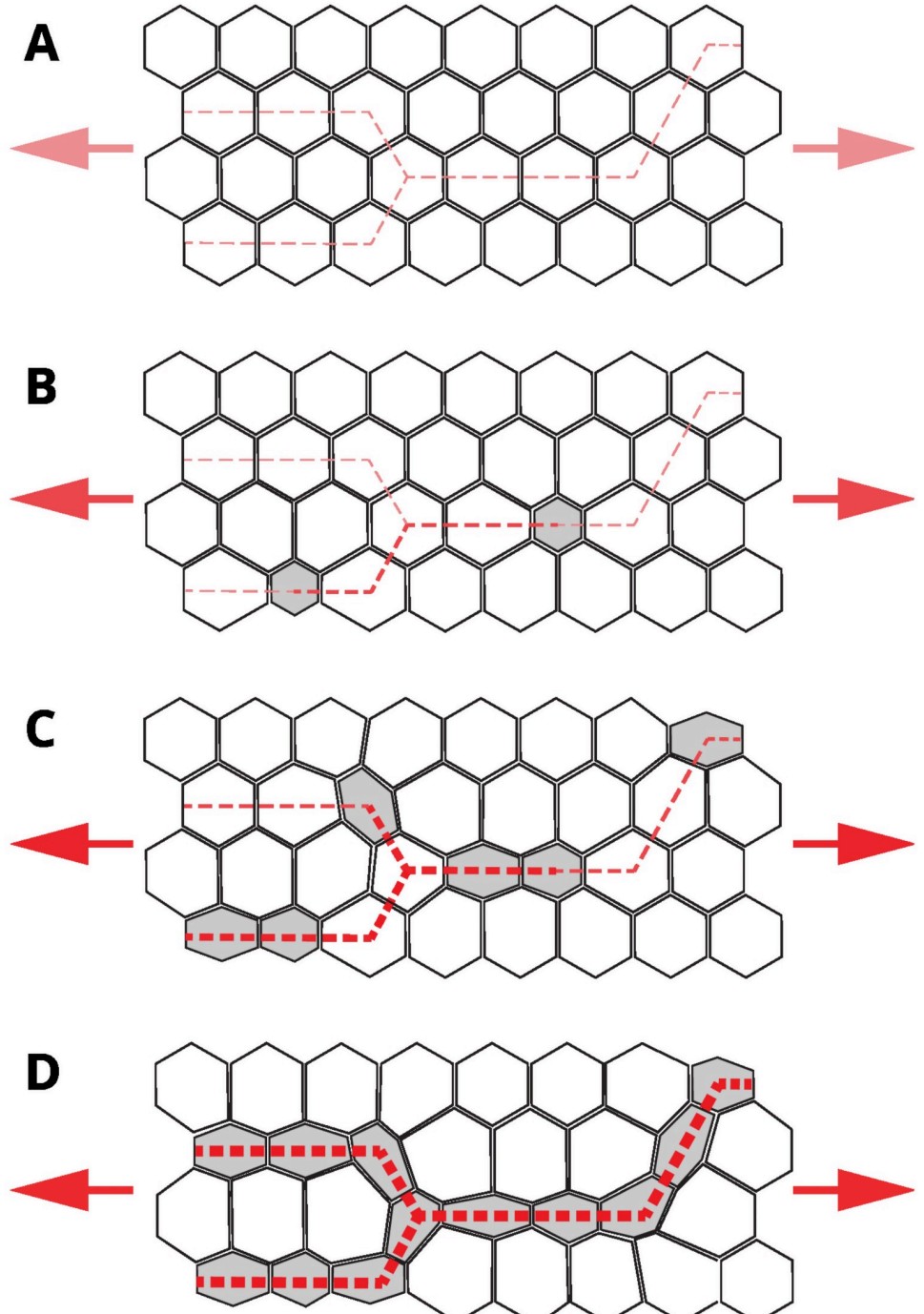

**Fig 15. Schematic model for tensile stress-propagated apical constrictions in the early ventral furrow primordium.** (**A**) The specified VF primordium is a mechanically active tissue under regional tensile stress from the adjacent anterior and posterior ends containing constricted cells (arrowheads, red). Example lines of tensile stress (dashed lines, red) in the tissue before apical constrictions begin. (**B**) Early ratcheted apical constrictions (gray) are distributed along the lines of tensile stress, but do not appear connected. As cell apices constrict, tensile stress increases along the stress lines (indicated by darker red coloration and greater thickness at dashed lines). (**C**) A later timepoint showing more apical constrictions (gray) forming along the lines of tensile stress. (**D**) The VF primordium before the fast phase of rapid apical constrictions initiated by the Fog and T48 signaling pathways. Apical constrictions are visible in chains.

Recent experimental results of Yevick *et al.* [26] show apically constricting cells forming chains around an ablated hole in the mesoderm primordium, consistent with both our experimental results and our simulations presented here. Moreover, at the lowest photoactivation level the embryo undergoes uninterrupted VF invagination that is only weakly disturbed by the local actomyosin contractility reduction. Experimental reduction of overall contractility across the VF primordium also showed only weak disruption of the process [26]. Based on this observation and our theoretical predictions we propose that mechanical feedback serves to increase robustness of the apical constriction process involved in VF invagination.

The robustness of VFF in response to experimental perturbation has been demonstrated in both the work of Guglielmi *et al.* [25], our analysis of it, and in other recent reports [16, 26]. Here, we show that CCC morphology and the associated stress distribution, both aided by mechanical feedback during the initial slow phase of apical constrictions, play an important role in achieving well organized and robust VFF.

## The role of mechanical feedback in other morphogenetic events

While our study focused on the role of tensile feedback during the slow phase of apical constrictions in VFF, other morphogenetic phenomena are expected to be mechanically coordinated in similar ways. Mechanical feedback is likely to be involved, for example, in convergent extension, which can act to simultaneously narrow a tissue in one dimension and lengthen it in another. In one example of convergent extension, *Drosophila* germband extension, cells intercalate through polarized actomyosin contraction and subsequent cellular interface shortening to exchange neighboring cells. The exchange can either take place in a single-cell–single-cell intercalation or through formation of polarized supracellular actomyosin cables. The cables subsequently constrict to generate multicellular rosettes, which resolve in the orthogonal direction to produce tissue elongation [44–47]. Like VFF apical constrictions, convergent extension might be coordinated by mechanical interactions. In particular, the contractility enhancement induced by tensile forces may result both in cable formation and coordinated cable contraction. Convergent extension is also involved in vertebrate development during gastrulation and neurulation. Convergent extension in these morphogenetic processes involves pulsed actomyosin contractions [48–51], so a similar feedback mechanism may be at play. Recently, it has been shown that blastopore closure in Xenopus is also aided by tension produced by convergent thickening [52, 53].

In another embryonic process in *Drosophila*, dorsal closure, epidermal tissue spreads over another tissue. This process involves apical constriction of the cells of the underlying amnioserosa and the contraction of a supracellular actomyosin cable in the spreading tissue (in addition to other mechanisms) [28]. The amnioserosa is under tension and cells undergo apical constrictions associated with pulsatile actomyosin contractions [54–56] similar to those involved in VFF. It is thus likely that tensile stress generated by the contraction pulses spatially organizes apical constrictions to aid dorsal closure.

Tissue level tension in *Drosophila* has been shown to affect the orientation of mitosis in cell division. This has been observed in certain groups of cells dividing in the extending germband [57]. Tension caused by supracellular actomyosin cables along the *Drosophila* embryonic parasegmental compartment boundaries orients cell division polarity [58]. Likewise, tension generated by actomyosin contractility overrides the general tendency for cell division primarily along the axis of tissue elongation in the follicle cells surrounding the oocyte [59].

Beyond a direct effect on the mechanical aspects of morphogenetic processes, mechanical deformation and mechanical feedback can regulate gene expression, which has been

demonstrated in VFF [11, 13]. There is also evidence that mechanical feedback regulates gene expression in a variety of other organisms [60].

In recent years it has become increasingly evident that mechanical-feedback control is omnipresent in embryonic development. Reverse-engineering of how genetic expression, chemical signaling, and mechanical feedback work together to achieve a robust formation of the intricate embryonic architecture requires not only targeted experiments but also predictive theoretical models of mechanically responsive active biological matter on both the local and the whole-embryo level. Our present study is a step in this direction. While the aim of our AGF approach in this work was to develop the simplest possible model, future development will include new elements such as pulsatile constrictions or dorsoventral constriction variability. Further experiments to disrupt actin, myosin, adherens junctions, and their regulatory proteins by genetic means, or by physical means such as laser cutting, would provide greater insight into mechanical feedback operating in developmental processes. We are currently working on modeling mechanical-feedback effects during the entire VFF process [61].

## Methods

### Experimental methods

**Immunofluorescence imaging.**    *Drosophila melanogaster* embryos were fixed using the heat/methanol protocol [62]. Fixed embryos were stained with mouse anti-Nrt (1:10, Developmental Studies Hybridoma Bank, DSHB), rabbit anti-Zip [63] and Hoechst dye. Secondary antibodies used were goat anti-mouse Alexa Fluor 488 (Invitrogen) and goat anti-rabbit Alexa Fluor 546 (Invitrogen). Embryos were mounted in Aquapolymount (Polysciences). Embryos imaged in the mid-sagittal plane were manually oriented [64], and embryos imaged in cross-section were manually sectioned [65]. Imaging was performed with a Nikon Ti-E microscope with an A1 confocal system. Images are shown in Fig 1.

**Acquiring images of live embryos.**    Flies and embryos were cultured at 22.5˚C. Embryos carrying the *Spider-GFP* transgene [27] were used to visualize the plasma membrane and cell shapes. Embryos were prepared as described previously [5, 21, 64, 66]. Embryos were hand-selected for the optimum stage (early stage 5 [67]) of development in halocarbon oil 27 (Sigma). The oil was removed by moving the embryos onto agar and then mounting them with glue in embryo chambers designed to avoid compression artifacts [5]. The ventral sides of the embryos were imaged using a 40x oil objective (NA 1.3) of a Nikon Ti-E microscope with an A1 confocal system using a pinhole size of 1.6 AU and 4x averaging. Images were collected in three planes separated by 1 μm, at 15 second intervals. Five embryos with optimal orientation to view the VF field were imaged [24, 68]. Images are shown in Figs 1–3, 6 and 7. All relevant experimental data are available from Dryad [69].

**Image processing.**    Time lapses of live embryos were loaded into the Embryo Development Geometry Explorer (EDGE) software package for segmentation and tracking [68]. EDGE uses a combination of MATLAB routines to distinguish cellular membranes and then identifies cells by drawing an overlay to segment individual frames. This segmentation is an automated process based on user entered variables; however, corrections of the resulting overlay are often necessary and must be done manually. Cells are then tracked using polygon matching by comparing relative centroid location and fractional area overlap between images.

After segmentation and tracking, data were extracted from EDGE in the form of matrices that map the pixels of each frame to individual cells. These pixel matrices were used to identify and tag constricted cells. Tracked cells were marked as constricted based on their minor axis length reduction relative to the reference value, as defined in Eq 1. The minor axis length was calculated for each cell in a given image from the pixel matrices by solving for the eigenvalues

of each cell's second moment matrix. Second moment matrix of cell *i* is defined by

$$
\boldsymbol{M}_i = \sum_j \begin{bmatrix} (p_{xj} - r_x)^2 & (p_{xj} - r_x)(p_{yj} - r_y) \\ (p_{xj} - r_x)(p_{yj} - r_y) & (p_{yj} - r_y)^2 \end{bmatrix},
\tag{16}
$$

where $(p_{xj}, p_{yj})$ are indices of pixel *j* contained in the discrete array of pixels that describes cell *i*, and $(r_x, r_y)$ are the indices of the pixel which coincides with the cell centroid. These eigenvalues effectively define the length of major and minor axes as though cellular geometries were projected onto an ellipse. Cell-specific reference minor-axis lengths were established by averaging measurements over 15 to 20 sequential frames prior to the onset of apical constrictions. Time zero was selected as the first frame in a given time lapse experiment that has a cell which remains constricted across multiple subsequent frames using a minor-axis reduction cutoff of 65% to identify constricted cells.

The pixel matrices are also used to generate the cellular neighbor list necessary for identifying clusters. Cells are identified as adjacent neighbors through an iterative process where each pixel of a given cell is checked to see if it is a primary or secondary neighbor to the pixel of any other cell. A list of constricted cell clusters for each frame is then generated by scanning the neighbor list to discern whether constricted cells are adjacent neighbors to any other constricted cells. The list of clusters allows for statistical measures such as total number of clusters and sizes of clusters to be monitored.

**Identification of active cells.** Active cells in the imaged portion of the VF mesoderm primordium are defined as those cells that experience minor axis reduction by a factor of at least $r = 0.85$ during the entire slower-phase constriction process. The number of active cells evaluated in this way is used to calculate the fraction of active cells that have constricted, $\%N_c$, and the normalized average number of clusters, $C_{avg}$.

## Theoretical AGF model

**Simulation domain.** In our coarse-grained approach the entire apical cell ends are represented as 2D mechanically coupled stress-responsive active particles that are capable of random constrictions, as described in the Results section. The curvature of the cellular layer is neglected, and the region of interest is represented as a planar domain (see Fig 4). To accurately represent the relevant domain of the embryo, our full-scale simulations are performed in a square domain with $N_{tot} = 6,400$ closely packed particles. This particle number approximately equals the number of cells in the *Drosophila* embryo during the VFF process.

**Boundary conditions.** We use the periodic boundary conditions both in the horizontal (anteroposterior) and vertical (dorsoventral) directions *x* and *y*. The periodicity in the dorsoventral direction *y* reflects the approximately cylindrical shape of the embryo. The embryo has anterior and posterior end caps that are relatively immobile throughout the VFF process. We use the periodic boundary condition in the anteroposterior direction *x* to approximate the relatively rigid boundary of the end caps while avoiding complexities associated with implementing specific boundary conditions for each cap. Alternatively, one could use fixed boundary conditions in the *x* direction; however, such approach would not be more realistic, because the immobile end caps are in high curvature areas of the embryo, which cannot be accurately represented using fixed boundary conditions in a rectangular flat domain.

**Particle size distribution.** To mimic polydispersity of *Drosophila* cells, the system in the initial state (i.e., before the cell constrictions occur) is a disordered 50%–50% mechanically stable mixture of particles with diameter ratio $r = 1.1$ [70], based on the repulsive-interaction range of the interparticle potential Eq 4.

**Preparation of the initial state.**   The initial state of the system is prepared using the algorithm described in [5]. We first generate a mechanically stable packing of $N_{tot}$ particles interacting via the repulsive spring potential. Subsequently, we establish the neighbor list based on the center-to-center criterion $r_{ij}/d_{ij} < 1.1$. The neighbor list defines connected neighbors, which interact via both the attractive and repulsive parts of the spring potential Eq 4. Since cell intercalation does not occur during the slow phase of VFF, the list remains unchanged throughout the simulation of the constriction process.

**System equilibration.**   The system is equilibrated after each constriction step by solving Newton's equations of motion for a system of particles interacting via the spring potentials Eq 4 and by additional interparticle dissipative forces [70]. The system is evolved using the standard velocity Verlet algorithm [41] until the system reaches the potential energy minimum.

**Implicit mesoderm representation.**   To lower the numerical cost we also use the system with a reduced number of inactive cells, in which only a portion of the inactive lateral region is modeled explicitly. The effect of the remaining inactive cells on the behavior of the system is approximated by elastic springs acting upon the cells at the border of the explicit domain (see Fig 4D). The spring constant matches the elastic properties of the replaced lateral domain, determined from full simulations of the entire system of 6,400 cells.

**Evaluation of the virial stress.**   The dimensionless virial stress tensor associated with the forces acting on particle $i$ is evaluated from the standard expression

$$S_{\alpha\beta}(i) = -\frac{1}{2\epsilon}\sum_{j\neq i} r_{ij\alpha} f_{ij\beta}, \tag{17}$$

where $\alpha, \beta = x, y$ indicate Cartesian components, $r_{ij\alpha} = r_{i\alpha} - r_{j\alpha}$ is the $\alpha$ component of the relative position of particles $i$ and $j$, $f_{ij\beta}$ is the $\beta$ component of the force exerted by particle $j$ on particle $i$, and the summation is over the interacting neighbors. The stress tensor defined by Eq 17 (Eq 10 in the absolute tensor notation) is symmetric, because the particles are torque-free. The major (minor) stress is the eigenvalue of the stress tensor with the larger (smaller) magnitude, and the major (minor) axis is the direction of the corresponding eigenvector.

**Constriction perturbation coefficient.**   The constriction perturbation coefficient Eq 15 is evaluated as the ratio of the average values of the fractions $\%N_c^t$ and $\%N_c^u$ of active cells that constricted in the affected and unaffected domains over $n$ independent simulation runs. We used $n = 120$ simulations for tensile sensitive system and $n = 16000$ for the random system.

The theoretical curves shown in Figs 10C, 12C and 12D are obtained from Eq 15 and the exponential-decay expressions

$$\%N_c^t = 1 - e^{-\alpha_r \tau}, \tag{18a}$$

$$\%N_c^u = 1 - e^{-\tau}, \tag{18b}$$

$$\%N_c = q(1 - e^{-\alpha_r \tau}) + (1-q)(1 - e^{-\tau}) \tag{18c}$$

where $\tau$ is the rescaled time, and $q$ is the fraction of active cells in the affected zone.

The constriction perturbation coefficients $\chi_{cp}$ and $\chi_{cp}^E$ are obtained by combining Eqs 15 and 18. For the bypass coefficient we have $\chi_{cp}^B = 1$, and for the combined coefficient $\chi_{cp}^{EB}$ we use

Eq 18 and

$$\chi_{\mathrm{cp}}^{\mathrm{EB}} = \frac{b\,\%N_{\mathrm{c}}^{\mathrm{t}} + (1-b)\,\%N_{\mathrm{c}}^{\mathrm{u}}}{\%N_{\mathrm{c}}^{\mathrm{u}}}, \tag{19}$$

where $b$ is the fraction of affected particles in the combined affected and bypass region.

**Dimensionless parameters of the model.** Our analysis focuses on geometrical characteristics of the constriction process which are independent of the dimensional quantities such as the average effective cellular diameter $d$ and the characteristic energy scale $\epsilon$. Represented in a dimensionless form, the results depend only on appropriate dimensionless combinations of relevant quantities and the associated coupling constants. For example, consider the stress feedback parameter $s$ defined as the dimensionless ratio between the current stress $\sigma_i$ and the reference stress $\sigma_{\mathrm{ref}}$ (see Eq 6). Parameter $s$ is independent of the dimensional quantities $\epsilon$ and $d$, because they cancel out; thus, $\epsilon$ and $d$ do not affect the constriction process. In what follows, we discuss all the dimensionless parameters of our model.

**Parameters based on the embryo geometry**. These parameters include the total number of particles $N_{\mathrm{tot}} = 6,400$; aspect ratio of the simulation domain $\lambda = 1$ (80 by 80 cells); and width of the active region $w_a = 12$ particles. In all our simulations the above parameters are unchanged.

**Other geometrical parameters**. The initial particle size distribution is defined by the diameter ratio $r = 1.1$ (the standard value used in particulate-media studies to prevent formation of ordered domains in random close-packed states [70]). The list of connected neighbors interacting via adhesive forces is determined using the cutoff $r_{ij}/d_{ij} = 1.1$.

**Energy-minimization parameters**. In our energy-minimization procedure we use a dimensionless damping coefficient $b = 0.5$. The system evolution is terminated when the relative potential energy changes by less than $10^{-3}$ within 1000 simulation steps. These parameters do not affect the simulation results but only control the numerical efficiency and resolution.

**Constriction factor**. Constricting particles reduce their diameter by the constriction factor $f_c$. In all simulations that are compared with the experimental data we use $f_c \approx r_c$, where $r_c$ is the constriction threshold used to identify constrictions of a given strength.

**Stress-feedback parameters**. The normalization coefficient $N_a$ in Eq 8 ensures that only a small number of particles constrict during a single simulation step. The specific value of this normalization coefficient does not influence the simulation results. The stress-sensitivity profile is described by the parameter $p$; the value $p = 3$ has been chosen to ensure that the constriction process is more sensitive to larger stresses generated by constricted cells and less sensitive to the fluctuating background stresses. This parameter is the same in all our simulations. The stress-coupling parameter $\beta$ describes the sensitivity of particle constrictions to tensile stress. This is the only parameter that is used to fit numerical results to the experimental data. The local probability amplitude $\alpha_i$ is used to define a disrupted region in the robustness study, but it does not affect any other results.

**Stress-scale factor in augmented Voronoi tessellation**. The scale factor $s_0$ defines the weight with which the virial stress affects the shape of Voronoi cells. The effect of this factor on the cell shape is small as long as $s_0 \approx 2$ (the approximate number of springs contributing to stress in a given direction). We use a fixed value $s_0 \approx 2.1$ to generate all Voronoi tessellation results presented in this study. The parameter $s_0$ does not affect the dynamics of the constriction process; therefore, the cluster counts presented in Fig 9 are independent of this parameter. The cluster orientations presented in S1 Appendix are determined taking into account the Voronoi cell shapes; the dependence of the orientation order parameter $\psi_2$ on the value of $s_0$ is very weak.

**Matching the simulations to the experimental data**. The stress-coupling parameter $\beta$ has been used as the only fitting parameter. The chosen value $\beta = 250$ gives the best overall agreement with experiments for all quantities shown in Fig 9, for clusters of both strongly and weakly constricted cells. The parameter $f_c$ was chosen based on the strength of the analyzed constrictions. The other parameters of the model had fixed values in all our simulations.

## Supporting information

**S1 Fig. Cellular constriction patterns for neighbor-triggered constrictions.** A developing constriction pattern is shown for a model system in which unconstricted active cells with $n_c$ constricted neighbors have their constriction probability increased by a factor $\beta_{cn} = 1 + 3.33 n_c$. Unlike tensile feedback, neighbor-driven constriction enhancement does not lead to formation of a connected network of constriction chains.
(TIFF)

**S2 Fig. Comparison of three different Voronoi tessellation approaches.** (**A**) The standard monodisperse tessellation assumes that all cells are size-weighted equally and draws membranes exactly halfway between cell centers, as described by the shape tensor $\mathbf{D}_i = \mathbf{I}$ in Eq 11. (**B**) The polydisperse tessellation moves the membranes closer to smaller cells and further from larger cells based on the ratio of their sizes ($\mathbf{D}_i = d_i \mathbf{I}$). (**C**) The stress-augmented tessellation weights membrane placement between two cells based on the mechanical stress that each cell experiences [$\mathbf{D}_i = d_i(\mathbf{I} + s_0^{-1}\mathbf{S}_i)$, where $\mathbf{S}$ is the virial stress, as described in more detail in the main text]. The stress-augmented Voronoi algorithm renders the most realistic representation of a confluent cellular medium with chains of constricted cells (brown).
(TIFF)

**S3 Fig. A comparison of the stress distribution between systems with and without tensile feedback.** In the system with tensile feedback (top) tensile stress is supported by chains of constricted cells, while unconstricted cells are subject to relatively small tensile forces. In the random uncorrelated system (bottom) there are large groups of unconstricted cells that bear strong tensile stress. Results are shown for a system with 40% of cells constricted; the color scheme and system parameters per Fig 8.
(TIFF)

**S4 Fig. Panels (F-G) from Fig 13 with manually marked constricted cells.** Constricted cells for the 1.5 mW case were manually identified based on overall size relative to other cells in frame and relative visual difference between the major and minor axis of each cell's individual shape. Scale bars: 10 μm. Figure modified and reprinted from [25] under the article's CC BY license.
(TIFF)

**S5 Fig. Panels (J-K) from Fig 13 with manually marked constricted cells in the photoactivated region.** Constricted cells for the 0.7 mW case were manually identified based on overall size relative to other cells in frame and relative visual difference between the major and minor axis of each cell's individual shape. Only the optogenetically affected area and the area immediately surrounding it were considered due to low contrast of the unaffected areas in (K). As the edge of the forming furrow can be seen in the left side of (K), this low contrast is likely the result of the unaffected areas beginning to invaginate. Scale bars: 10 μm. Figure modified and reprinted from [25] under the article's CC BY license.
(TIFF)

**S6 Fig. Representative control embryo.** The apical surface of the ventral mesoderm of a wild-type control embryo expressing only CIBN::pmGFP. Frames show the embryo (**A**) 10 min before, (**B**) 5 min before, and (**C**) at the onset of ventral furrow formation. Scale bars: 10 μm. Figure reprinted from [25] under the article's CC BY license.
(TIFF)

**S1 Appendix. Orientation of constricted-cell clusters.**
(PDF)

## Acknowledgments

We would like to show our gratitude to Dr. Stefano De Renzis for sharing confocal microscopy images of optogenetically perturbed embryos. We would also like to thank Dr. Hannah G. Yevick for useful discussions at the 2018 APS March Meeting.

## Author Contributions

**Conceptualization:** Michael C. Holcomb, Guo-Jie Jason Gao, Jeffrey H. Thomas, Jerzy Blawzdziewicz.

**Formal analysis:** Michael C. Holcomb, Dylan Schneider, Jerzy Blawzdziewicz.

**Investigation:** Michael C. Holcomb, Guo-Jie Jason Gao, Mahsa Servati, Dylan Schneider, Presley K. McNeely, Jeffrey H. Thomas, Jerzy Blawzdziewicz.

**Methodology:** Michael C. Holcomb, Guo-Jie Jason Gao, Jeffrey H. Thomas, Jerzy Blawzdziewicz.

**Project administration:** Jeffrey H. Thomas, Jerzy Blawzdziewicz.

**Resources:** Jeffrey H. Thomas, Jerzy Blawzdziewicz.

**Software:** Michael C. Holcomb, Guo-Jie Jason Gao, Dylan Schneider, Jerzy Blawzdziewicz.

**Supervision:** Michael C. Holcomb, Jeffrey H. Thomas, Jerzy Blawzdziewicz.

**Validation:** Michael C. Holcomb, Guo-Jie Jason Gao, Dylan Schneider, Jeffrey H. Thomas, Jerzy Blawzdziewicz.

**Visualization:** Michael C. Holcomb, Guo-Jie Jason Gao, Mahsa Servati, Jeffrey H. Thomas, Jerzy Blawzdziewicz.

**Writing – original draft:** Michael C. Holcomb, Jeffrey H. Thomas, Jerzy Blawzdziewicz.

**Writing – review & editing:** Michael C. Holcomb, Guo-Jie Jason Gao, Mahsa Servati, Dylan Schneider, Jeffrey H. Thomas, Jerzy Blawzdziewicz.

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
