## [Decision Letter · Decision Letter 0]

9 Oct 2020

Dear Dr. Blawzdziewicz,

Thank you very much for submitting your manuscript "Mechanical feedback and robustness of apical constrictions in Drosophila embryo ventral furrow formation" for consideration at PLOS Computational Biology.

As with all papers reviewed by the journal, your manuscript was reviewed by members of the editorial board and by several independent reviewers. In light of the reviews (below this email), we would like to invite the resubmission of a significantly-revised version that takes into account the reviewers' comments.

We cannot make any decision about publication until we have seen the revised manuscript and your response to the reviewers' comments. Your revised manuscript is also likely to be sent to reviewers for further evaluation.

Sincerely,

Philip K Maini

Associate Editor

PLOS Computational Biology

Mark Alber

Deputy Editor

PLOS Computational Biology

Reviewer's Responses to Questions

**Comments to the Authors:**

Reviewer #1: In this paper, Holcomb et al investigate processes preceding ventral furrow formation in Drosophila, where cells constrict apically before the furrow starts forming. Specifically, the authors consider the spatial coordination of constricting cells. The authors have previously observed that constricting cells in this system form long, connected chains that span in the antero-posterior direction of the embryo. How this spatial alignment occurs mechanistically has not been clear. Here, the authors combine computational modelling and image analysis to show that mechanical feedback can explain the spatial alignment of constricting cells that has been observed experimentally. Intriguingly the model captures many qualitative features of these chains. For example, biologically chains of constricting cells grow ‘towards each other’, which in the model is explained nicely through the mechanical feedback, and the model can quantitatively fit distributions of chain lengths over time. This new explanation of spatial coordination is supplemented with an intriguing analysis of the model, which illustrates that such mechanical feedback can support developmental robustness. In a system where cells constrict due to mechanical feedback, constriction chains can ‘invade’ or surround regions in which contractility is reduced, leading to a larger fraction of constricting cells than in a scenario where mechanical feedback is absent. These latter results tie in nicely with previously published data from optogenetic experimental perturbations of contractility, and the authors discuss the similarities between their model and these existing data.

I really enjoyed reading this thoroughly conducted and extensive paper. The paper informs multiple topics that are currently important in developmental biology research, namely the investigation of spatial cross-talk between cells on short lengthscales, the role and control of developmental mechanics, as well as the emergence of developmental robustness. This is a timely paper that I believe will make a strong contribution to Plos Computational Biology.

I only have a few minor comments that I hope the authors can clarify before publication.

1) How is the model parameterised? Specifically, which model parameters where fitted in figure 9? What were the values of the remaining model parameters and how were these identified?

2) Also for figure 9: In the fitting, is the model including feedback fitted to the data, and feedback then turned off to produce the curve without feedback, or are both scenarios somehow fitted to the data individually? I’d be worried that the former might introduce bias to the model comparison.

2) In your model, chains form always in antero-posterior direction. Is this simply a result of the imposed boundary conditions, i.e. the symmetry of the domain, or is there another, more explicit bias to introduce this anisotropy?

3) Figure 3: After having stared at this for a while, I believe I can see the similarities you mention in the text, but then maybe I am only imagining things because I have looked at it for so long. Is there any way to make this more explicit? Maybe similarities can be highlighted with arrows or some such? Or perhaps it’s possible to quantify the similarities as the proportion of cells at each time point that have already been counted as constricted at a previous timepoint, under a weaker cutoff? Also, I don’t understand why the top right panels are not displayed.

3) On page 9, it is not clear to me why the potential between cells is defined as a piecewise function, since both pieces seem to have the same functional form. Is it so to clarify that only the attractive part of the potential is restricted to interactions with nearest neighbours? Since the range for the repulsive contribution is smaller than the range of the attractive contribution, won’t the repulsive contribution only act on nearest neighbours anyway?

4) Page 9, line 163: ‘The normalisation by N_A ensures that approximately the same number of cells constrict in each time step.’ Why is this important, is this the case biologically? Does the propensity to constrict increase for each cell as the process progresses in the real tissue? Can you comment on possible mechanisms for this?

5) Page 3, line 42 “… that these data are inconsistent with results for purely random uncorrelated constrictions, or for only neighbour-correlated constrictions” – Which part of the paper do you refer to with ‘only neighbour-correlated constrictions’? This was not clear to me.

6) Page 6, line 116 “… a typical vertex model likely would be inadequate for description of this system, whereas a particle-based model that treats cells as undivided entities described by their effective properties is more appropriate” – I agree that a vertex model would introduce complexity that is not necessary for your model, and that the particle based approach is better suited here. I don’t think a vertex model would be inadequate though, I am quite convinced that a vertex model could well reproduce your results! Maybe rephrase this? Also, sorry for this comment, I use vertex models regularly, so I am obviously biased.

7) Is it possible to add colorbars to the figures where saturation denotes stress magnitude, such as in figure 8?

8) In Figure 13, can you indicate constriction chains in some way? It is difficult for me to see where constriction chains are in these images.

9) Will your simulation code be available somewhere? (I believe that is journal policy for Plos Comp Biol)

Reviewer #2: See attached PDF

Reviewer #3: The spatio-temporal organization of stress and deformation during a Drosophila embryo’s ventral furrow formation is an important topic in current developmental biology. The manuscript by Holcomb et al. centers on the application of theoretical models from the physics of granular fluids to identify the role of mechanical regulation of cellular behavior – namely, the onset of apical constriction. In previous work, the authors developed a computational model of the ventral furrow region in which the chain-like organization of constricting cells (as seen in experiments) occurred when local stress directly promoted contraction. In the current manuscript, this model is used to make a number of quantitative predictions which are directly verified with experimental data. The computational model is shown to successfully predict the size distribution of constricting clusters within the VF region, and further the authors demonstrate that stress-based feedback promotes cluster formation in the presences of local contractility perturbations. While I have some specific comments and suggestions, I overall recommend this paper for publication.

Comments:

1) The authors note that they choose to use their AGF model instead of a more commonly used vertex model. I think the model they use is reasonable, especially in light of the experimental data, but I would like to see a bit of expansion of the paragraph addressing the differences between said models. In particular, I do not entirely understand the claim: “A typical vertex model likely would be inadequate for description of this system, whereas a particle-based model that treats cells as undivided entities described by their effective properties is more appropriate.” Is there a reason to think that a vertex model under similar boundary conditions, and with similar stress-based feedback, would not exhibit contractile chains?

2) The stress-modified Voronoi tessellation the authors develop make intuitive sense to me, given significant role of stress in determining the shape of cells in vivo. However, I believe it would strengthen the paper to introduce a quantitative basis for the statement “We find that the augmented Voronoi algorithm renders a realistic representation of a confluent cellular medium with a significant degree of polydispersity and local anisotropy associated with the presence of CCCs”. For instance, if you take the cell positions from Fig. 2 and assuming the stress can be calculated from these positions using the AGF force terms, how much deviation is there between the Voronoi calculated cell shapes and real ones? I also think a cartoon illustrating the tessellation calculation might be helpful if added to Fig. 4.

3) It seems like particles keep the same neighbors over the course of each simulation, but it would be good to clarify this point if true. The lack of T1 transitions and other cell rearrangement is an important feature of tissue dynamics at this state, so if this is not being enforced it should at least be commented on.

4) Fig. 9 is reasonably convincing evidence, but it would be useful to have additional metrics for comparison between experiment and simulation, either in the main text or in the SI. For instance, it would be helpful to see a direct quantitative comparison at this point of the AP directional bias of the clusters for the cases presented in Fig 9.

5) Visually, it would be good to include a horizontal dotted line denoting a_r in Fig. 10 C.

6) Generally, attempts to demonstrate robustness involve demonstrating successful mechanical behavior in the context of plausible perturbations to the in vivo system. The discussion of robustness is mainly carried out in relation to an optogenetically controlled mutant embryo, which represents a more contrived sort of perturbation than a simple genetic change. As I read it, the authors argue that their experimental data should be taken as representative of a general class of mutations which undermines contractility in a local region of cells. To better shore up this point, it might be good for the authors to include references to other mutants which exhibit regions of low expression in the ventral furrow region.

7) The modification of the model in the robustness section reduces the contraction probability of particles, but intuitively I would expect the reduction of myosin recruitment observed in the would also influence the magnitude of the contractions observed in constricting cells. The authors should comment on this, as I would assume that if the computational model incorporated reduced contraction in the perturbed region, the corresponding stress and thus stress feedback effect in this region would likewise be diminished.

Reviewer #4: Holcomb et al present a largely model-based paper, but with some image quantification from published work, arguing that cell constriction during ventral furrow formation in Drosophila involves mechanical feedback, resulting in chains of constricted/ing cells.

This is a very interesting and topical area, so I was very interested to review this paper. It is clear, nicely presented and I found it easy to read, but in its current form I found it unconvincing.

Major points

A feature that worried me throughout the whole paper was that in vivo, cells nearest the ventral midline have the earliest and strongest apical Myosin activity (Xie & Martin) and start pulsatile and ratchetted constriction first. This activity subsequently spreads away from ventral towards the lateral edge of the mesoderm domain. So the effective width of the domain of contracting cells is rather narrower than the 12 cells the authors use in their simulations. From the in vivo data in Figs 1,2,3 & 7, the active domain that shows DV constriction over the developmental epoch presented is around 6 cells wide, and rather than showing separated chains of contracted cells, it is more like a block of cells, with many connections that are not along AP. This does not really agree with the patterns of simulated chains that emerge, for example, in Fig 8.

Some detail is shown in Figure 7 of chains supposedly emerging in vivo. The central chain was plausible but I could not understand how chains could branch off in DV-oriented directions when tension is supposed to be along AP. So, given that this is just an illustrative example, the proof should be in the summary of quantified in vivo and simulated data in Fig 9. Here, to be convinced of tension propagation, I would need to see results for simulations that focus only on the most ventral 6 rows of cells, using etas and %Nc for just this region, not including all the non-chaining lateral cells, which would mean a much more dense field of constricting cells. In each of the left hand graphs (A,C,E,G), I could see the pink star line (no mech feedback) moving towards the purple triangle line (mech feedback) in this scenario. So I am not yet convinced that the data support mechanical feedback. The right hand graphs (for weaker constrictions) clearly do not favour mech feedback.

I think it’s also true to say that the evidence in Xie and Martin (2016) for chains of constricting cells is quite weak, and that the orientation of chains that they show are not strongly biased to be AP-oriented. So I would downgrade this evidence to being suggestive of the possibility of mechanical feedback, but biochemical communication (via extracellular fog or via more direct cell-cell signalling) or entrainment of biochemical oscillators or… are also possible explanations. Surely if tension chains were real and as clear as the authors suggest, that paper would have found much stronger evidence? It’s possible of course that there is some weak tension propagation, but I think the tools you would need to disentangle this would be local mechanical perturbation in vivo, cutting across single ‘chains’ with laser ablation for example.

The authors seem to be claiming that in vivo cell contractility (in DV typically) is a pure readout of stress, whereas surely the effective material stiffness can also vary, and locally? What would the implications be for their model and for the interpretation of cell shapes of locally varying effective cell stiffness? I would have thought that contractility (negative strain) by some cells in DV would help other nearby cells also contract in DV (cell adhesion is strong and cells don’t slide, as the authors note) in order to maintain tissue integrity.

When it came to the later sections of the paper on the function of tensile feedback, I’m not sure I yet buy the premise nor agree with the evidence. From the Martin et al (2009) paper, apical constriction of the most ventral cells happens remarkably quickly, over around 6 ratcheted pulsatile contractions. I’m not sure this is long enough to really benefit from repair through tensile feedback, and the more relevant feature that ensures robust morphogenesis is the spatiotemporal gradient of actomyosin activation from ventral-most cells first through to more lateral cells later. Would it not be a worry that mechanically propagated permanent tensile chains would be a dangerous positively reinforcing runaway process? Though there is no consensus yet on the function of cell pulsatility (hence this submitted paper), but it may be there precisely to ensure that chains are ephemeral/short-lived, to ensure coordinated tissue contractility that does not ‘runaway’.

Because actomyosin contractility would be the means through which tensile chains would be set up, I found it difficult to imagine an in vivo scenario where actomyosin contractility was sufficiently impaired locally that it was knocked out, but then able to be rescued by mechanical induction. Maybe the separate roles of Twist and Snail in the ratchet and generating fluctuations would be a candidate. Adam Martin’s group have shown that they can start cell fluctuations through physical perturbation when they are absent, but that is not really equivalent to showing tensile feedback of the kind required for chains.

For the Fig 13 de Renzis panels, I didn’t really see any chains in the 0.7 mW panels J & K, nor did I understand how the authors explain, according to their tensile chains idea, how the normal tissue in Fig. 13H, either side of the mid-section, becomes quite perturbed, with some larger and some smaller cells like the mid-section. Their quantification of the ‘chains’ in Fig 14A & B shows that it’s a network, without any long oriented chains. Shouldn’t there be a trough of rescued chains rather than a peak in the centre of the graph in Fig 14B (top panel), as seen in the more extreme bottom panel? The patchwork of contracted cells amongst unusually big cells does not seem to me to fit with the tensile chains hypothesis.

Overall, the authors still have quite a lot to do to convince that tensile stress chains are a feature of ventral furrow formation.

Minor points

Lines 158-161 are crucial for following the thrust of the paper and should be introduced much earlier.

Line 207, I didn’t understand ‘polydispersity’

I found the paper long-winded in sections covering Figs 10-14. Salient features could be brought out more succinctly.

**Have all data underlying the figures and results presented in the manuscript been provided?**

Reviewer #1: **No: **I could not find any supplementary data with this submission. Specifically, the software code does not seem to be available.

Reviewer #2: Yes

Reviewer #3: Yes

Reviewer #4: Yes

PLOS authors have the option to publish the peer review history of their article (what does this mean?). If published, this will include your full peer review and any attached files.

Reviewer #1: No

Reviewer #2: No

Reviewer #3: No

Reviewer #4: No
---

## [Decision Letter · Decision Letter 1]

6 May 2021

Dear Dr. Blawzdziewicz,

Thank you very much for submitting your manuscript "Mechanical feedback and robustness of apical constrictions in Drosophila embryo ventral furrow formation" for consideration at PLOS Computational Biology. As with all papers reviewed by the journal, your manuscript was reviewed by members of the editorial board and by several independent reviewers. The reviewers appreciated the attention to an important topic. Based on the reviews, we are likely to accept this manuscript for publication, providing that you modify the manuscript according to the review recommendations.

Sincerely,

Philip K Maini

Associate Editor

PLOS Computational Biology

Mark Alber

Deputy Editor

PLOS Computational Biology

[LINK]

Reviewer's Responses to Questions

**Comments to the Authors:**

Reviewer #1: The authors have addressed all my comments very clearly and thoroughly. As I described in my previous review, I think this is an excellent paper that would make a strong contribution to Plos Computational Biology. I hence recommend it for publication.

One little thing I noticed that I would hope can be addressed without further revisions is the following: The authors now explain the parameterisation of the model and the fitting very clearly in the paper. However, there is a parameter called epsilon in cell-to-cell force potential that appears to have been left out of these explanations. In the revised paper, I could not find the value that was chosen for this parameter epsilon and why it was chosen. Perhaps the authors can add a simple one or two sentences about this parameter in the final publication. In any case, I would assume that this parameter only controls the characteristic timescales of the dynamics, not the geometrical effects analysed in the paper.

Reviewer #2: I thank the authors for the substantial work they have provided, which improves a lot their manuscript. I have a few more comments below:

FIG13: I maintain that the paper should be self-contained as far as possible and that the control from Guglielmi et al should be added to the figure (or as a Supp fig).

L181: Mathematically this is an affine relation (not linear)

Reviewer #3: The authors have thoroughly addressed my comments from the previous review, and I recommend this revised version for publication.

Reviewer #4: I use the same numbering system the authors have used in their ‘Response to Major Points’.

Points 1 & 3: relating to the 6 vs 12 cell wide domain. The central argument of the paper hinges on Figure 9 which the authors claim shows that their model with tensile feedback closely matches in vivo data. Prior to Figure 9, the comparisons between model and in vivo data are visual, but in Figure 9 the comparison becomes objectively quantitative. I raised the issue in my review that because constricting cells in vivo are mostly packed into a central 6 cell wide tranche of the tissue, rather than throughout the 12 cell wide domain of Twist the authors have replicated in their model, would this change the interpretation of Figure 9? I thank the authors for repeating their model for the 6 cell wide domain, but unless I have missed something, for comparison to data they should adjust %Nc for the in vivo data too (in Fig 1-R4). %Nc is the “fraction of active constricted cells”, the number of constricted cells divided by the total number of cells in the active region. Since the latter denominator has halved in the 6 cell domain, the in vivo data should also be recalculated with the (approx.) halved denominator. This will have the effect of stretching the in vivo data to the right in the 3 eta panels in Fig 1-R4. For the the first C(ave) panel, there is a double effect on the denominator of C(ave) as well as on the %Nc x-axis. Could the authors please make this adjustment to see how the 6 cell wide model compares to 6 cell wide in vivo data? This should then be incorporated into Figure 9 and the conclusions adjusted as necessary.

If the conclusions of Fig 9 still hold as the authors currently have them, or if my reasoning above is awry, then I am happy with the authors’ responses to my other points.

**Have the authors made all data and (if applicable) computational code underlying the findings in their manuscript fully available?**

Reviewer #1: Yes

Reviewer #2: Yes

Reviewer #3: None

Reviewer #4: Yes

PLOS authors have the option to publish the peer review history of their article (what does this mean?). If published, this will include your full peer review and any attached files.

Reviewer #1: No

Reviewer #2: No

Reviewer #3: No

Reviewer #4: No

Figure Files:

Data Requirements:

Reproducibility:

References:

---

## [Editor Report · Decision Letter 2]

10 Jun 2021

Dear Dr. Blawzdziewicz,

We are pleased to inform you that your manuscript 'Mechanical feedback and robustness of apical constrictions in Drosophila embryo ventral furrow formation' has been provisionally accepted for publication in PLOS Computational Biology.

Best regards,

Philip K Maini

Associate Editor

PLOS Computational Biology

Mark Alber

Deputy Editor

PLOS Computational Biology

---

## [Editor Report · Acceptance letter]

30 Jun 2021

PCOMPBIOL-D-20-01470R2 

Mechanical feedback and robustness of apical constrictions in Drosophila embryo ventral furrow formation

Dear Dr Blawzdziewicz,

I am pleased to inform you that your manuscript has been formally accepted for publication in PLOS Computational Biology. Your manuscript is now with our production department and you will be notified of the publication date in due course.

With kind regards,

Zsofi Zombor
